

# Influence of gravity wave temperature anomaly and its vertical gradient on cirrus clouds in the tropical tropopause layer – a satellite-based view

Kai-Wei Chang[1] and Tristan L'Ecuyer[2]

[1]Department of Atmospheric Sciences, Texas A&M University, College Station, Texas, USA
[2]Department of Atmospheric and Oceanic Sciences, University of Wisconsin-Madison, Madison, Wisconsin, USA

**Correspondence:** Kai-Wei Chang (kchang37@wisc.edu)

**Abstract.**

Negative temperature perturbations ($T'$) from gravity waves are known to favor tropical tropopause layer (TTL) clouds, and recent studies have further suggested a possible role of $dT'/dz$ on facilitating TTL cloud formation and maintenance. With a focus on exploring the influence of $dT'/dz$ on TTL clouds, this study utilizes radio occultation temperature retrievals and
cloud detection from the Cloud-Aerosol Lidar and Infrared Pathfinder Satellite Observations (CALIPSO) to understand how gravity wave perturbations modulate cloud occurrence.

Cloud populations were evaluated in four phases corresponding to positive or negative $T'$ and $dT'/dz$. We find that 57% of TTL clouds are found where $T'$ and $dT'/dz$ are both negative. Regions of frequent convection are associated with higher cloud populations in the warm phase $T' > 0$. The partitioning of cloud population among wave phases shows some dependence on
the background relative humidity estimated using Aura Microwave Limb Sounder water vapor retrievals. Using effective radius ($r_e$) retrievals from the CloudSat/CALIPSO 2C-ICE product, we find that $r_e$ is distributed similarly among all wave phases but a smaller mode is found in the $r_e$ distribution from the phase $T' < 0$ and $dT'/dz < 0$.

It is shown that the strongest mean negative $T'$ anomaly is centered on the cloud top, resulting in positive $dT'/dz$ above the cloud top and negative $dT'/dz$ below. This negative $T'$ anomaly propagates downward with time consistent with upward
propagating gravity waves. Negative (positive) $T'$ anomalies are associated with increased (decreased) probability of being occupied by clouds. The magnitude of $T'$ correlates with the increase or decrease in cloud occurrence, giving evidence that the wave amplitude influences the probability of cloud occurrence. While the decrease of cloud occurrence in the warm phase is centered on the altitude of $T'$ maxima, we show that the increase of cloud occurrence around $T'$ minima occurs below the minima in height, indicating that cloud formation or maintenance is facilitated mainly inside negative $dT'/dz$. Together with
existing studies, our results suggest that the cold phase of gravity waves favor TTL clouds mainly through the region of wave anomalies where $dT'/dz$ is negative.





## 1 Introduction

Variations in stratospheric water vapor (SWV) influence the rate of surface warming due to climate change (Solomon et al.,
2010) and have a significant climate feedback (Banerjee et al., 2019). There is a need to better understand mechanisms controlling the amount of SWV, as they are potentially key components of climate change and stratosphere-troposphere coupling. Since the large-scale slow upwelling throughout the tropical tropopause layer (TTL) brings air from the troposphere into the lower stratosphere, conditions and processes in the TTL modulate the amount of water vapor in the lower tropical stratosphere. Cirrus formation by cold temperatures in the TTL is generally regarded as the primary mechanism dehydrating air entering
the stratosphere (Holton et al., 1995). Studies have shown that cirrus cloud occurrence strongly associates with Kelvin waves (Immler et al., 2007; Fujiwara et al., 2009) and gravity waves (Suzuki et al., 2013; Kim et al., 2016), and that these waves can enhance the dehydration occurring inside the TTL (Schoeberl et al., 2015).

Previous studies on waves and cirrus clouds generally show that enhanced cirrus cloud occurrence tend to coincide with the gravity wave phases with negative temperature anomalies. Through aircraft observations of the NASA Airborne Tropical
TRopopause EXperiment (ATTREX) campaign (Jensen et al., 2013), Kim et al. (2016) (K16 hereafter) show that ice was found most frequently where the temperature anomaly ($T'$) and vertical slope of temperature anomalies ($dT'/dz$) were both negative, bringing the latter quantity into attention as a possible control on cirrus formation. Since K16 showed that the occurrence of convectively-coupled clouds had no preference towards the sign of $T'$ or $dT'/dz$, the tendency of TTL clouds to occur in negative $T'$ and $dT'/dz$ likely depicts a connection between clouds and gravity wave perturbations. They suggest that
the negative rate of change in temperature (positive cooling rate) in regions where $dT'/dz < 0$, due to the downward phase propagation of gravity waves, may facilitate cloud formation and explain the abundance of cloud in the phase with $T' < 0$ and $dT'/dz < 0$. Another explanation of high cloud frequency in this phase is given by Podglajen et al. (2018) (P18 hereafter) who used a simplified set of equations to model the interaction between ice crystal growth, sedimentation, and gravity wave perturbations of temperature and vertical motion. P18 argues that in this phase the upward vertical motion acts in concert
with the sedimentation rate of crystals with certain sizes, suspending crystals inside this wave phase as it descends with the downward phase propagation of the wave. Motivated by these studies, we aim to further explore this connection between gravity waves and cirrus clouds through satellite datasets.

This study utilizes temperature profiles from the radio occultation (RO) technique (Kursinski et al., 1997) which has been widely used to study equatorial gravity and Kelvin waves (Randel and Wu, 2005; Alexander et al., 2008; Scherllin-Pirscher
et al., 2017). We collocate these RO profiles to cirrus cloud observations from the Cloud-Aerosol Lidar and Infrared Pathfinder Satellite Observations (Winker et al., 2010) to study the relationship between gravity/Kelvin wave phases and cirrus occurrence. Especially within 2006 to 2014 there is a high spatial and temporal density of RO soundings from the U.S.-Taiwan joint mission Constellation Observing System for Meteorology, Ionosphere, and Climate (COSMIC) (Anthes et al., 2008), allowing a large number of RO profiles to be collocated with cirrus cloud retrievals.

Collocations that are temporally close in time are used to evaluate wave perturbations $T'$ and $dT'/dz$ in relation to cloud occurrence and properties. In addition, since the sampling of COSMIC is pseudo-random in both space and time, it is possible to





obtain RO profiles that are spatially close to CALIPSO footprints but before or after the time of the footprint. Using collocations of various time separations we build composite time series of wave anomalies and cloud frequency to understand how waves are influencing TTL clouds. Finally, we use the Aura Microwave Limb Sounder (MLS) water vapor retrievals (Read et al.,

2007) and ice cloud effective radius ($r_e$) retrievals from the CloudSat/CALIPSO 2C-ICE product (Deng et al., 2013, 2015) to evaluate whether relative humidity and $r_e$ have connections to gravity waves as shown by P18.

Section 2 describes the datasets used in this study. In Section 3 we explain the extraction of wave temperature anomalies from RO profiles, and the method for data collocation. In Section 4, results are given in three parts: Section 4.1 discusses the population of clouds in each wave phase, Section 4.2 presents the composite time evolution of wave anomalies and cloud

frequecy, and Section 4.3 evaluates the predictions of P18 with satellite observations. Our conclusions are summarized in Section 5.

## 2 Satellite products

This study uses the re-processed radio occultation (RO) atmPrf dataset processed by the Cosmic Data Analysis and Archive Center (CDAAC). We use occultations from the following satellite missions: Constellation Observation System for Meteorol-

ogy, Ionosphere, and Climate (COSMIC) (Anthes et al., 2008), Meteorological Operational Polar Satellite A/Global Navigation Satellite System Receiver for Atmospheric Sounding (Metop-A/GRAS) (Von Engeln et al., 2009), Metop-B/GRAS, and the Challenging Minisatellite Payload (CHAMP) (Wickert et al., 2001). Because the RO technique does not suffer from inter-satellite calibration effects (Foelsche et al., 2011), profiles from different satellite missions can be used together as long as they are processed with the same algorithm. The level 2 atmPrf dataset provides 'dry' profiles of atmospheric temperature

derived by neglecting moisture, which is appropriate for TTL altitudes. The atmPrf provides temperature estimates at 30-meter vertical spacing, but the effective resolution of RO is around 200 meters in the tropical tropopause layer (Zeng et al., 2019). The precision of temperature is approximately 0.5 K within 8 to 20 km (Anthes et al., 2008).

The Cloud-Aerosol Lidar and Infrared Pathfinder Satellite Observations (CALIPSO) (Winker et al., 2010) is a sun-synchronus, polar-orbiting satellite along the NASA A-Train formation, overpassing the equator at 0130 and 1330 local solar time. Its pri-

mary instrument, the Cloud-Aerosol Lidar with Orthogonal Polarization (CALIOP), is a dual-wavelength lidar capable of detecting subvisual clouds with optical depths less than 0.01. We use the Level 2 V4.10 5-km Cloud Layer product for estimates of cloud top and base altitude and the V4.10 5-km Cloud Profile product for detection of clouds in 60-meter vertical bins. The Cloud Aerosol Discrimination (CAD) Score in these products is a measure of confidence that the detected feature is correctly classified as cloud. To ensure high confidence that all analyzed features are clouds, our analysis only includes

CALIPSO layers and bins with CAD score at or above 80 (where 100 means complete confidence in the feature being a cloud). Corresponding to the dates when RO data were available, we use nighttime CALIPSO data between 2006 and 2014. Daytime CALIPSO data were excluded due to the lower signal-to-noise ratio of daytime CALIOP observations.

Estimates of ice cloud effective radius ($r_e$) come from the 2C-ICE product (Deng et al., 2013, 2015) which is derived jointly using the CloudSat radar and CALIPSO lidar observations. This product provides $r_e$ retrievals at 1-km footprints in 250-m



vertical bins. The $r_e$ estimates from 2C-ICE compare well to in-situ flight measurements with a retrieval-to-flight ratio of 1.05 (Deng et al., 2013). For quality control, we only use $r_e$ with uncertainty (given by the re_uncertainty variable) less than 20%. Due to a battery failure CloudSat left the A-Train formation in 2011. After that it only operated in daytime and its footprint was no longer collocated to CALIPSO. For this reason we limit our analysis to 2C-ICE from 2007 to 2010 when nighttime data was available.

The Aura MLS $H_2O$ product provides retrievals of water vapor mixing ratio at pressures at and below 316 hPa with a precision of 0.2-0.3 ppmv (4-9%) in the stratosphere (Lambert et al., 2007). We use the water vapor mixing ratio to estimate relative humidity with respect to ice ($RH_i$) using collocated RO temperature. Criteria for data screening follows all the recommendations outlined in section 3.9 of the official documentation (Livesey et al., 2017) found at https://mls.jpl.nasa.gov/data/v4-2_ data_quality_document.pdf. Although Aura was launched in 2004, the scan of the MLS did not align with that of CALIOP

until May 2008. For this reason, all analysis involving this product uses data from 2008 to 2014.

## 3   Methods

### 3.1   Gravity wave temperature anomalies

Our method for obtaining temperature perturbations ($T'$) due to gravity waves is based on Alexander et al. (2008). Mean temperature profiles are calculated on grid boxes of 20° longitude × 5° latitude × 7 days centered on each day of year. Mean

maps are made for each day between 1 Nov 2006 and 30 April 2014 during which COSMIC provided a large number of RO observations. For an arbitrary RO temperature profile, the mean map centered on the same day as the RO profile is used to derive the corresponding mean-state profile through bilinear interpolation of the four grid boxes surrounding the location of the given RO profile. $T'$ is then obtained by removing the mean-state from the actual profile. Since we use a 7-day mean state, the resulting $T'$ can be thought of as representing variability on timescales less than seven days. After $T'$ is obtained, its vertical

gradient is calculated to get $dT'/dz$. Figure 1 shows one example of a temperature profile, its corresponding mean state, and the resulting anomalies $T'$ and $dT'/dz$.

### 3.2   Collocation of CALIPSO observations to RO profiles

The primary goal of this work is to study cirrus occurrence and properties in the four gravity wave phases defined in Figure 1. To accomplish this we collocate CALIPSO cloud observations to RO temperature profiles. The horizontal weighting of

RO retrievals is mostly centered within 200 to 300 km of the perigee (tangent) point (Kursinski et al., 1997) where the ray experiences most bending. For this reason we use the spatial location of the perigee point as basis for collocation. Since our interest lies strictly inside the TTL and the perigee point of each occultation ray changes with height, we determine the perigee point at the middle of the TTL by interpolating the longitude and latitude of RO profiles to 17.25 km (middle of TTL determined as the average of 14.5 km and 20 km). Any CALIPSO observations within 100 km of this point are collocated to the RO profile

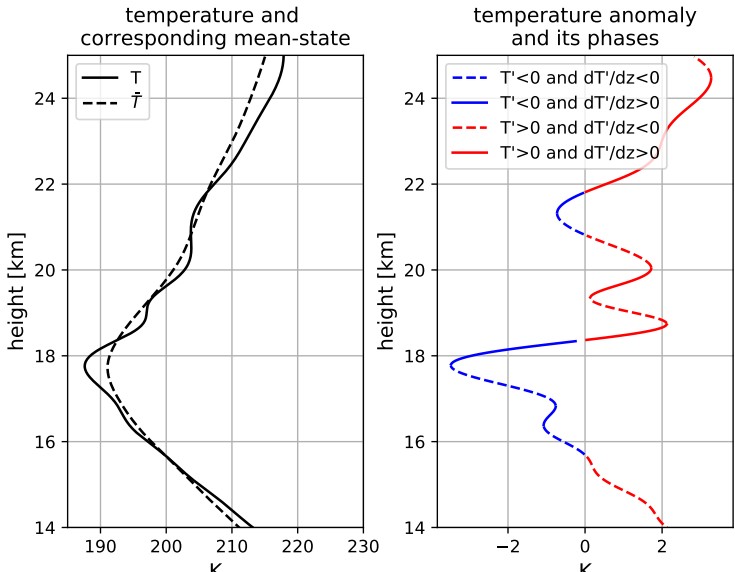

**Figure 1.** (Left) Temperature profile (solid line) from COSMIC at $155° \, 43'$ W, $18° \, 16'$ N on 1 Jan 2007 and its corresponding mean-state (dashed). (Right) $T'$ of the given profile and its four phases based on the sign of $T'$ and $dT'/dz$.

for analysis. Figure 2 gives an example of one RO profile, its perigee point at 17.25 km, and the collocated CALIPSO 5-km footprints.

We collocate RO profiles to 2C-ICE cloud retrievals in a similar manner. Unlike the CALIPSO 5-km products, 2C-ICE provides cloud properties at 1-km footprints and vertical bins of approximately 250 m. Other than this difference, the collocation method is identical to that of CALIPSO and RO. In May 2008 the Aura MLS was aligned to within $\pm$ 10 km of CALIOP. For analysis involving $RH_i$, for each CALIPSO footprint with a RO collocation, we find the closest MLS footprint to that CALIPSO footprint to calculate $RH_i$.

## 4 Results

All results below were derived from data within $20°$ of the equator. For convenience we will refer the the four gravity wave phases as follows. Phase 1: $T' < 0$ and $dT'/dz < 0$, Phase 2: $T' < 0$ and $dT'/dz > 0$, Phase 3: $T' > 0$ and $dT'/dz < 0$, and Phase 4: $T' > 0$ and $dT'/dz > 0$. Cold and warm phases refer to where $T' < 0$ and $T' > 0$, respectively.

For the analysis presented in Section 4.1 and Section 4.3, the temporal restriction for collocation is that all the collocated data must be within two hours of each other. This restriction is not imposed in 4.2, and the time difference between RO and CALIPSO observations range from 0 to 36 hours with the purpose of examining how waves and cirrus clouds tend to evolve over time. This will be further elaborated in that section.





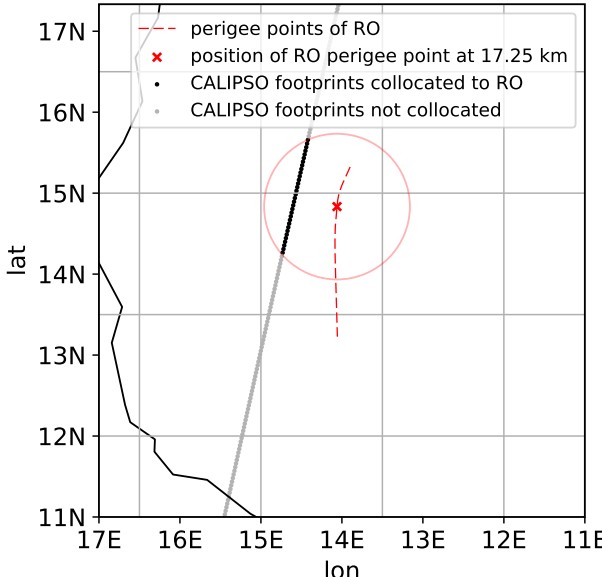

**Figure 2.** Schematic of collocation between RO profile and CALIPSO footprints. The perigee points of the RO profile throughout all altitudes are shown in the red dashed line, while the position of the perigee point in the middle of the TTL (17.25 km) is denoted by the red X. The CALIPSO 5-km product provides estimates of cloud properties at 5-km footprints, and all footprints within 100 km of the red X are collocated with the RO profile, as indicated by the black dots. Gray dots are CALIPSO footprints are considered too far from the RO profile and not collocated. The shown RO profile was taken at approximately 0120UTC 2 Jan 2009.

## 4.1 Population of clouds in wave phases

As previously mentioned, K16 (their Figure 5) found that a majority of TTL clouds in the ATTREX data were observed in the cold phase $T' < 0$ and that in the 2014 flight legs over the Western Pacific there was a higher frequency of ice inside $dT'/dz < 0$ than in $dT'/dz > 0$. To assess whether this tendency is general throughout the TTL or is limited to the regions observed by ATTREX, we evaluate cloud populations using collocated CALIPSO and RO observations that cover the TTL at all longitudes and over 2007–2014. Figure 3 shows the population of CALIPSO Cloud Profile vertical bins detected as clouds in each wave phase extracted from collocated RO profiles. Considering all collocated observations between 1 Nov 2006 and 30 April 2014, 57% of clouds are observed to occur in Phase 1 throughout the entire TTL, as shown in Figure 3(a). When the cloud population is examined in 1-km vertical layers (14.5–15.5 km, 15.5–16.5 km, etc.), there is no obvious change with height and most clouds are found in Phase 1 followed by Phase 2 at all heights. Above 16.5 km there is a smaller fraction of clouds in the warm phase. A possible explanation for this may be that there are less convectively detrained clouds as altitude increases, increasing the probability of clouds having been formed by gravity waves. In addition, the population in Phase 2 tends to increase with height, with 38% of clouds above 17.5 km in Phase 2. For comparison, using K16's Figure 5 one can infer that for clouds above 16.5 km the cloud fraction in Phase 1, 2, 3, and 4 are 56.25%, 31.25%, 9.375%, and 3.125% (calculated





as the percentage in that phase divided by the sum all four phases), and for clouds below 16 km the percentages are 49.30%,
28.17%, 14.08%, and 8.45%. These ratios are similar to our findings, though we find less clouds in Phase 2 below 16 km.

Cloud fractions in each phase are separated into six longitudinal belts in Figure 4. During December-January-February
(DJF), the 120°E-180°E belt, which covers the Maritime Continent and Western Pacific, has the lowest cloud population in
Phase 1 (52%) as well as the most clouds inside the warm phase (24%). Although this region is known for very low tropopause
temperatures and high TTL cirrus frequency during boreal winter (Highwood and Hoskins, 1998; Sassen et al., 2009), there
is also frequent deep convection (Ramage, 1968) which may generate clouds unrelated to gravity waves. This may explain
the higher cloud population in warm phases. The influence of convection may also explain the high warm phase population
during June-July-August (JJA) in 60°E-120°E where there is frequent convection due to the Asian Monsoon. In this period
and region, 28% of clouds are in the warm phase and 47% is in Phase 1.

Over 180°W-120°W, which approximately covers the Eastern Pacific, 62% (27%) of clouds fall in Phase 1 (Phase 2) during
DJF . This is in contrast with K16 who found that in the 2011 and 2013 ATTREX flight legs over the Eastern Pacific there were
more clouds in Phase 2 than in Phase 1. The 2011 flights were conducted in October and November while the 2013 flights
were in February and March. Plots similar to Figure 4 made from data in October-November (not shown) yielded Phase 1/2
populations of 63%/24% while February-March yielded 58%/20%. Over this region we were not able to find Phase 2 having
more clouds as K16 did. It is not clear what causes this difference. Their $T'$ were calculated as the difference between aircraft
in-situ temperature and 30-day mean temperature derived from RO, while we calculate it as the difference between RO-derived
temperatures and 7-day mean profiles. This difference in methodology may be a factor causing the contrasting findings.

Using the cloud top and base heights reported from the CALIPSO Cloud Layer product we calculate the vertical cloud
fraction in each wave phase, defined as the amount of vertical overlap between the cloud boundaries and the line segments in
Figure 1 corresponding to each phase. The distributions of vertical cloud fraction are shown in Figure 5 (for cloud fractions
between 0 and 1) and Table 1 (for cloud fractions equal to 0 and 1). In this figure and table we only consider clouds with base
above 14.5 km and wave phase segments whose base height lie within 14.5 and 18.5 km. In phases of positive $dT'/dz$, the
number of samples tend to decrease as cloud fraction increases. This trend doesn't apply for negative $dT'/dz$, and in Phase
1 there is a clear increase of samples with increasing cloud fraction. Phase 1 also has the most cases where the vertical cloud
fraction is unity (19410 cases). Overall, Phase 1 is most distinct as it has most samples with cloud fractions of unity and it
significantly favors higher values of vertical cloud fraction.

**Table 1.** Number of cases with zero or unity vertical cloud fraction (CF).

| Wave Phase | CF=0 | CF=1 |
| --- | --- | --- |
| Phase 1 | 432,838 | 19,410 |
| Phase 2 | 461,251 | 10,656 |
| Phase 3 | 441,906 | 9,583 |
| Phase 4 | 457,166 | 999 |





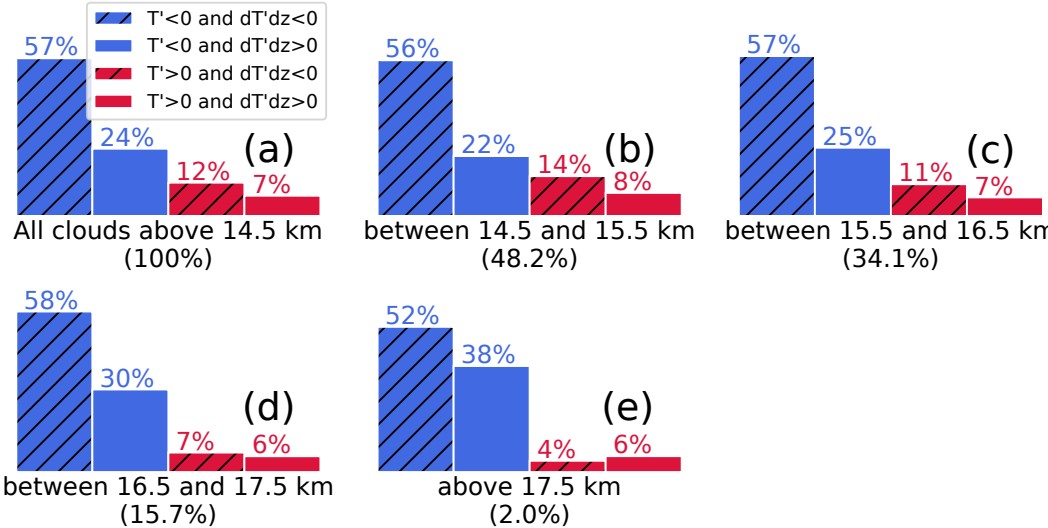

**Figure 3.** Population of CALIPSO Cloud Profile cloud bins inside each wave phase from 1 November 2006 to 30 April 2014. Plot (a) shows the cloud fraction for all of TTL (14.5 to 20 km), while (b)–(e) show the population in three different 1-km layers. The percentage in parenthesis denote the portion of clouds found in that vertical layer relative to all TTL clouds.

## 4.2 Composite time evolution of wave anomalies and cirrus occurrence

Since COSMIC observations are pseudo-random in time and space, it is possible to collocate CALIPSO observations to RO soundings with varying offsets in time. By binning the temperature profiles according to the time offsets, we can make a composite showing the mean time evolution of wave anomalies relative to the cloud observation. Such an approach of creating composite time series has been used to study the thermodynamic budget before and after tropical convection (Masunaga, 2012; Masunaga and L'Ecuyer, 2014), temperature anomalies associated with tropical deep convection (Paulik and Birner, 2012), and the interaction between atmospheric dust and tropical convection (Sauter et al., 2019).

We bin RO profiles in time bins of $-35, -33, ..., -1, 1, ..., 33, 35$ hours relative to the CALIPSO observations, where a negative value indicates that the collocated RO profiles precede the CALIPSO overpass. The composites of $T'$, $dT'/dz$, and buoyancy frequency $N^2$ anomaly for all collocations in 2006 to 2014 are shown in Figure 6. In making these composites we only includes clouds with cloud base of at least 14.5 km to ensure that the included clouds are TTL clouds (instead of, for example, convection) Also, for statistical testing, we need the RO profiles used in each time bin to be unique. For this reason, only the CALIPSO footprint spatially closest to the RO profile is used. If this is not done, then the same RO profile may be reused several times since there are usually multiple CALIPSO footprints collocated to one RO profile as shown in Figure 2.

In Figure 6(a), the strongest cold anomaly is found close to the cloud top and is coldest near hour 0. The cold anomaly contour with value below $-0.6$ K lasts approximately from $-15$ to $+6$ hours, and migrates downward with time consistent with the property of gravity and Kelvin waves with upward group velocity. The alternating cold-warm anomaly at heights of 2





**Figure 4.** Same as Figure 3(a) except for different longitudinal belts. Plots (a)-(f) are for December-January-February and (g)-(l) are for June-July-August.

to 6 km should be due to the diurnal tide (Zeng et al., 2008; Pirscher et al., 2010) since we are compositing only on nighttime data from CALIPSO which always overpasses the equatorial region at similar local times. The number of samples in each time



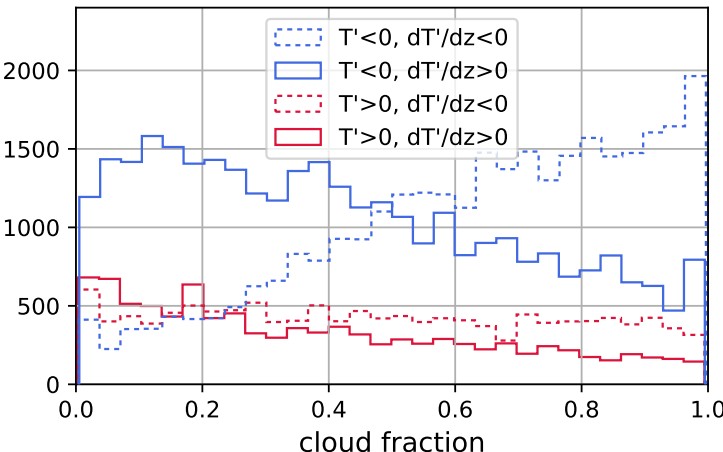

**Figure 5.** Distribution of vertical cloud fraction in each gravity wave phase. Blue and red colored lines indicate the cold and warm phase, respectively, while the solid and dashed lines represent $dT'/dz > 0$ and $dT'/dz < 0$. The number of cases with vertical cloud fractions of zero or unity are not plotted in this figure and is instead shown in Table 1.

bin (Figure 6(d)) has a 12-hour periodicity mainly due to Metop-A/B. When only using COSMIC observations to make Figure 6(a), (b),and (d), the anomaly patterns are largely the same so the periodicity does not affect the composites.

It is noteworthy that the mean cold anomaly is centered near the cloud top and not within the cloud. This results in a dipole structure in $dT'/dz$ (Figure 6(b)) and buoyancy frequency anomaly (Figure 6(d)) with positive anomalies just above the cloud top and negative anomalies below. This structure shows that the inside of clouds (below cloud top) is likely to have

negative $dT'/dz$, consistent with the finding by K16 and Figure 3 that a majority of clouds are found in Phase 1. Although this structure implies weakened stability (negative $N^2$ anomaly) inside the cloud, it is unclear whether this decreased stability has connections to cloud formation or maintenance. Since negative $dT'/dz$ also corresponds to upward vertical motion anomalies (assuming that these anomalies are from gravity waves), further study is required to separate the role of vertical motion and stability in how gravity and Kelvin waves influence TTL clouds.

Figure 7 is similar to Figure 6 except the anomalies are not composited relative to the cloud top height but rather on height above mean sea level. In this composite, there are cold anomalies at TTL altitudes but the magnitude is weaker than that of Figure 6. This leads to weak anomalies in $dT'/dz$ (Figure 7(b)) and $N^2$ (not shown). Based on these results we suggest that gravity wave anomalies in Figure 6 are physically significant and have a close association with the vertical position of TTL cloud tops.

P18 argues that the upward vertical velocity in Phase 1 slows down the descent of ice crystal and tends to suspend them inside Phase 1. Since these composites depicts a downward propagation of wave anomalies, it is of interest to investigate whether the phase propagation of gravity waves is associated with a downward migration of clouds. We can explore this possibility through a similar compositing technique employed above. Instead of centering on the CALIPSO footprint, we centered on the time





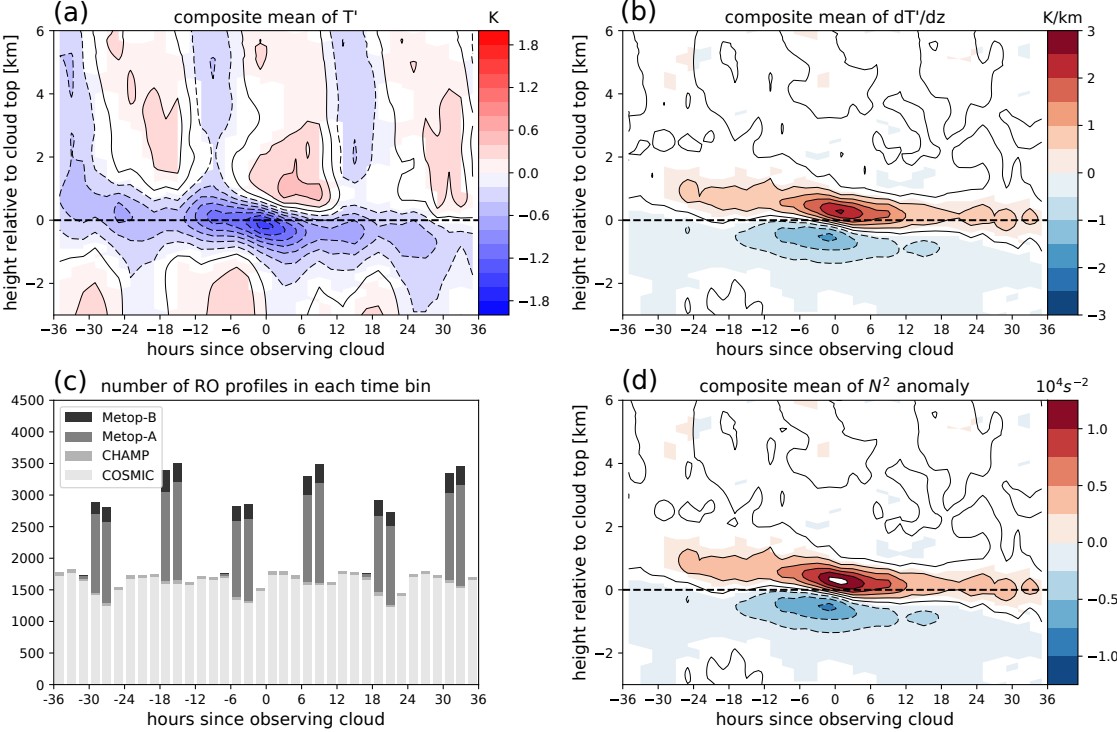

**Figure 6.** Composite of (a) $T'$, (b) $dT'/dz$, and (d) buoyancy frequency $N^2$ anomaly in height coordinate relative to cloud top. Colored contours in these three plots are at or above the 95% significance level according to the Student's t-test. Solid (dashed) contours represent positive (negative) anomalies and are at the same levels as the colored contours. The abscissa denotes the time offset between the CALIPSO observation and RO sounding. Plot (c) shows the number of unique RO profiles in each 2-hour time bin used to calculate the composites.

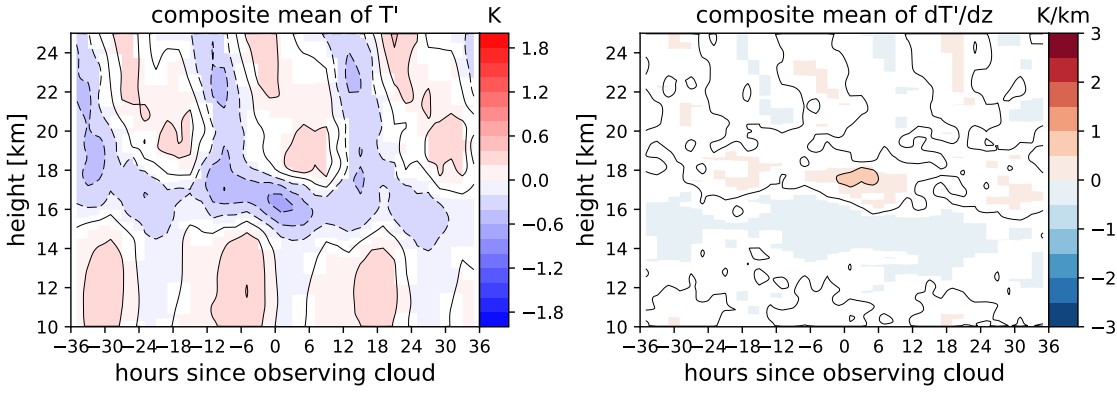

**Figure 7.** Same as Figure 6(a) and 6(b), except the ordinate denotes height above mean sea level. Colored contours are at or above the 95% significance level according to the Student's t-test. Contour levels are identical to Figure 6.





of RO sounding, and use the CALIPSO cloud product to calculate the cloud frequency in each 2-hour time bin. In addition,
instead of compositing relative to cloud top height, we composite on the altitude of the *local* minima or maxima of $T'$. A
schematic of this compositing approach is given in Figure 8.

In the example shown in Figure 8(a), at day $i$ there is a collocated RO sounding that occurred within 100 km of the CALIPSO
footprint but $\Delta t_i$ hours after. The position of the cloud top and base (solid and dashed magenta lines) is evaluated relative to
a local $T'$ minimum. Since the CALIPSO overpass occurred before this RO profile, in the compositing (shown in Figure 8(b))
the observed cloud position is used to calculate the cloud fraction at $\Delta t_i$ hours *before* the RO sounding. The cloud fractions are
calculated on a grid of 50-m height and 2-hour time bins. For the collocation pair in Figure 8(a), the cloud fraction in the time
bin corresponding to $t = -\Delta t_i$ is calculated according to how much each vertical bin overlaps with the interval $[h_i^{(b)}, h_i^{(t)}]$.
If the collocated CALIPSO footprint has no clouds with base above 14.5 km, cloud fractions of zero are still binned in the
appropriate time bin at all heights. Since any RO profile most likely has multiple local minima, the binning of cloud fraction is
repeated for each local minimum in a $T'$ profile. The exact same procedure is conducted for local maxima to create a separate
cloud frequency composite. To focus on TTL clouds we only include clouds with base at or above 14.5 km. Also, we only
consider local $T'$ extrema within 14.5 to 18.5 km since a majority of TTL clouds are inside this height range.

The composites of cloud frequency made this way, shown in Figure 9(a)-(c) and (d)-(f), can then be interpreted as the prob-
ability of finding clouds in the vicinity of local $T'$ minima or maxima, respectively. The shown values represent the mean
cloud fraction in each time-height bin. Each column is produced from a different subset of $T'$ extrema based on magnitude
($|T'| > 0.5$, 1.0, or 1.5 K). Beneath the vertical position of $min(T')$ (Figure 9(a)-(c)) we find a lobe of enhanced cloud fre-
quency and this becomes more evident as $min(T')$ decreases. Likewise, in the vicinity of $max(T')$ (Figure 9(d)-(f)) the cloud
frequency is reduced, and this reduction also shows dependence on the magnitude of $max(T')$. In both cases, the increased
or decreased cloud frequency displays a downward trend consistent with the expectation that gravity wave phases propagate
downward with time. However, it is hard to quantitatively know which parts of these plots are true anomalies. For this reason
we devise a way for extracting the anomalies in these patterns and for statistical testing, as described below.

The cloud frequency composites in the top two rows of Figure 9 are made using the altitude of the $T'$ extrema as the zero
height. To generate a composite where the vertical positon of $T'$ extrema has no relationship with cloud top/base height, for each
$T'$ extremum we generate a random altitude using uniform distribution $unif(14.5, 18.5)$ and make a separate cloud frequency
time-height composite with the random altitude as zero height, shown in Figure 9(g)-(i). These plots can be interpreted as the
cloud frequency distribution one would expect if the vertical position of $T'$ extrema has no connection to cloud top/base height.
Then the statistically significant differences between the first/second row to the third row may signify a connection between
wave anomalies and cloud occurrence. The distribution of cloud fraction in each time-height bin is similar to those shown in
Figure 5 and therefore is not normal, so the Student's t-test cannot be used here. We use the two-sided two-sample Kolmogorov-
Smirnov test (K-S test) (Hollander et al., 2015) which does not make any assumptions about the data distributions. This test
can be used to evaluate whether two continuous or discrete probability distributions differ from each other. The K-S test is
employed to compare the discrete cloud fraction distribution in each time-height bin of the first/second row to the same bin





in the third row. The null hypothesis is that the first/second row is not different than the randomly generated cloud frequency pattern in the third row.

The fourth row of Figure 9 (panels (j)-(l)) is the first row minus the third row, depicting the anomalies associated with $min(T')$, and similarly the fifth row (panels (m)-(o)) shows the anomalies associated with $max(T')$. Colored portions of the contour denote regions with $p < 0.05$ (95% confidence) as estimated from the K-S test. In these anomaly patterns it is confirmed that there is enhanced cloud occurrence below $min(T')$, and, in addition, a weak reduction of cloud occurrence above it. For the subset of $min(T') < -0.5$, the positive cloud frequency anomaly peaks at 3% whereas for $min(T') < -1.5$ it peaks at

6%. The anomaly patters due to $max(T')$ also exhibit a dipole structure with negative anomalies centered on the altitude of $max(T')$ and a weak positive anomaly below. The fifth row also suggests a dependence of cloud frequency anomaly with respect to the magnitude of $max(T')$, although the variation is not as large compared to that of $min(T')$. Both positive/negative anomalies associated with $min(T')/max(T')$ tend to migrate downward in time, although this trend is slightly more apparent in the enhanced cloud occurrence of $min(T')$.

One difference between $min(T')$ and $max(T')$ is that the positive anomalies in $min(T')$ occur below the altitude of $min(T')$ while the negative anomalies in $max(T')$ are centered on it. Most of the enhanced cloud occurrence occurs inside Phase 1, and Phase 2 actually tends to have a negative cloud occurrence anomaly. Although the predictions of P18 suggests that it may be more likely to find clouds in Phase 2 under low background $RH_i$, this global analysis suggests that on average the role of Phase 1 in facilitating TTL clouds is dominant.

## 265   4.3   Comparison to P18

P18 suggests that (1) ice crystals within a confined range of $r_e$ are suspended approximately in Phase 1 and 2, and (2) background relative humidity with respect to ice ($RH_{ib}$) influences the phase at which these crystals are suspended. These two features are depicted in their Figure 2. To evaluate whether these predictions are consistent with satellite observations, we examine $r_e$ and $RH_{ib}$ in observations to see whether these quantities exhibit any correlations to gravity waves. Although here

we present analysis motivated by P18, we note that their study assumes no background wind nor shear in their derivations and simulations.

    Figure 10 shows normalized distributions of $r_e$ in the four wave phases as well as their mean and standard deviation. These distributions only contain nighttime 2C-ICE data, since the information toward thin cirrus are mostly from lidar backscatter. Clouds above 17.5 km were omitted in this plot due to the low samples (∼0.6% of all TTL clouds). The distributions for all

phases are very similar regardless of height. In 14.5–15.5 and 15.5–16.5 km, the $r_e$ distribution of Phase 1 has a peak near 16 $\mu$m. Above 16.5 km this peak is not evident, but the Phase 1 distribution has higher values around 15 $\mu$m and lower values between 20 to 25 $\mu$m, slightly differentiating Phase 1 from the other phases. The mean $r_e$ of Phase 1 is lower than all other three phases at all vertical layers, but the differences are small. In summary, characteristics of $r_e$ found here are qualitatively consistent with P18's findings, as Phase 1 tends to have a relative larger number of ice particles localized around a certain $r_e$

value. However we note that retrieving cloud properties of thin cirrus has large uncertainties and more research is needed to explore the $r_e$ distribution in gravity waves phases using a variety of observations and models.





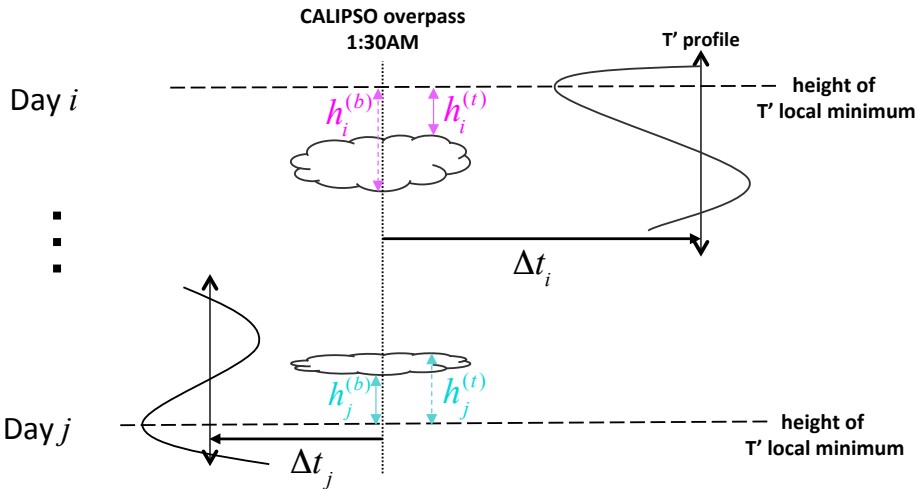

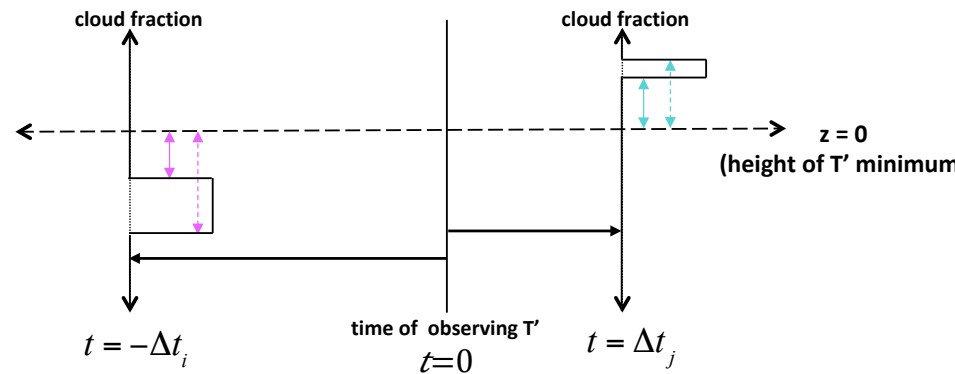

**Figure 8.** Schematic for creating the composite temporal evolution of cloud fraction with respect to $T'$ minima. In this example, as shown in (a), during day $i$ there is a collocated pair of CALIPSO and RO observations. The RO observation occurs $\Delta t_i$ hours after that of CALIPSO, and the vertical distance between the height of local $T'$ minimum and the cloud top and base is depicted by the dashed and solid magenta lines, respectively, with lengths $h_i^{(t)}$ and $h_i^{(b)}$. In the temporal compositing (b), in the time bin corresponding to $\Delta t_i$ hours before observing $T'$, the cloud fraction is binned according to how much each vertical bin overlaps with the interval $[h_i^{(b)}, h_i^{(t)}]$. The same procedure is carried out for the collocated pair at day $j$. Also see text for explanation.

As discussed in Section 4.1, K16 found that in the 2011 and 2013 flight legs over the Eastern Pacific there were slightly more clouds in Phase 2 than Phase 1, whereas in the 2014 flights over the Western Pacific a majority of clouds were in Phase 1. P18 argues that this may be due to the relatively low $RH_{ib}$ characteristic of the TTL over the Eastern Pacific. P18 solved a simplified

set of equations describing the interaction of gravity wave perturbations and ice particle growth/sedimentation. Comparison





of the solution using values of $RH_{ib} = 0.85$ or $0.64$ (to represent Western and Eastern Pacific, respectively) showed that the former results in the ice crystals being suspended in Phase 1 where in the latter ice particles were situated closer to the $T'$ minimum which may result in more ice inside Phase 2. Motivated by these results we collocate the MLS water vapor retrieval to CALISPO and RO data to evaluate whether observations suggest a similar dependence on $RH_{ib}$.

For each CALIPSO Cloud Profile bin identified as cloud, the water vapor mixing ratio from the Aura MLS product is log-interpolated (as suggested by the product documentation) to the height of the cloud bin. To evaluate the saturation mixing ratio, we interpolate the *7-day mean temperature* to the cloud height since we are interested in the $RH_{ib}$ instead of the actual $RH_i$ (which would include wave influence). The Goff-Gratch equation (Goff and Gratch, 1946) is used to get the saturation vapor pressure, and subsequently the saturation mixing ratio and $RH_{ib}$. Figure 11 shows the cloud population in each phase partitioned by $RH_{ib}$ values. The subsets with $RH_{ib}$ below 60%, between 60% to 80%, and above 180% tend to have less clouds in Phase 1 compared to the other $RH_{ib}$ categories. The intermediate values of $RH_{ib}$ (between 80% to 120%) yielded the highest fractions in Phase 1. Qualitatively, this show some consistency with P18 since there are more clouds in Phase 2 for the lowest two $RH_{ib}$ categories, but we find only a slight dependence on $RH_{ib}$ with no clear trend. We conclude that our analyses on $r_e$ and $RH_{ib}$ does not prove or disprove P18's assertions but there is some qualitative consistency between P18 and our results.

## 5   Conclusions

This study uses multiple satellite datasets to evaluate the influence of gravity wave perturbations on TTL cirrus clouds. With a focus on understanding the role of $dT'/dz$, the vertical gradient of the gravity wave temperature perturbation $T'$, we extract $T'$ and $dT'/dz$ from RO observations and collocate them to clouds observed by CALIPSO and 2C-ICE to understand cloud occurrence and characteristics relative to wave anomalies. Similar to the results of K16, we find that the phase where $T'$ and $dT'/dz$ are both negative (Phase 1) is most frequently occupied by TTL clouds. The second most populous phase is where $T' < 0$ and $dT'/dz > 0$ (Phase 2), followed by where $T' > 0$ and $dT'/dz < 0$ (Phase 3) and then $T' > 0$ and $dT'/dz > 0$ (Phase 4). We show that this relation among the four phases is more or less invariant with height or longitude.

A mean view of the temporal evolution of wave anomalies with respect to clouds is constructed by taking advantage of RO's pseudo-random distribution in time and space. We collocate CALIPSO cloud observations to RO soundings that occur before and after the CALIPSO observation, and by averaging a large number of observations with different time separations, a composite time series of wave anomalies is presented. These composites show that, on average, the strongest cold anomaly due to gravity waves tends to be centered on the height of cloud top, and this cold anomaly descends with time consistent with the downward phase propagation of gravity and Kelvin waves having upward group velocity.

In the cloud frequency composites made with respect to local $T'$ minima or maxima, we find that the decrease of cloud probability in the warm phase does not show clear dependence on the sign of $dT'/dz$. This is distinct from the cold phase, where cloud probability is increased mainly below $min(T')$ where $dT'/dz$ is negative. Together with existing studies, this result adds support to the idea that Phase 1 facilitates cloud formation and/or maintenance. Although the downward migration





of the increased cloud frequency may be due to ice sedimentation, this is unlikely to be the case for the decreased cloud
frequency associated with the warm phase. Hence the downward migration of increased/decreased cloud frequency in the
temporal composites is most likely due to waves with downward phase propagation. We also show that the positive or negative
cloud frequency anomalies strengthen with increasing magnitude of $T'$ minima or maxima, giving evidence on a global scale
that the wave amplitude is connected to the probability of cloud occurrence.

Finally, using satellite estimates of $r_e$ from 2C-ICE we assess the predictions of P18 which implies that one may observe a
narrower distribution ice crystal effective radius inside Phase 1. Their conclusion that the background relative humidity with
respect to ice affects the vertical position of clouds is also evaluated here by using $RH_{ib}$ based on the Aura MLS $H_2O$ product.
Among all phases, $r_e$ are distributed similarly but the distribution of Phase 1 had a notably sharper peak and than the other
three phases and also a slightly smaller mean $r_e$. The partitioning of cloud population among the four phases showed some
variation with different values of $RH_{ib}$, with Phase 1 having less clouds at very low or very high $RH_{ib}$, but no clear trend is
identified. Thus, while our satellite-based analysis has some qualitative consistency with the results of P18, it is insufficient for
verifying their assertions.

This study adds to the exist findings showing that Phase 1 has a distinct connection to TTL clouds. The findings of K16,
based on aircraft data limited to specific regions and time span, have been extended by our study which shows that the large
amount of clouds in Phase 1 is a general characteristic of the TTL. Based on our composite analysis using satellite data
spanning eight years (2007–2014), the connection between wave anomalies and cloud occurrence is evident: cold anomalies
are associated with the position of cloud top, and $T'$ amplitudes influence the increase or decrease in cloud frequency. The
purpose of constructing composite temporal evolution by piecing together collocated temperature and cloud observations is an
attempt to study processes occurring on a timescale typically unobserved by satellites. Although the resulting composites are
not true time series, the anomalies patterns depict signatures consistent with wave propagation and enhances our understanding
of how waves are connected to TTL clouds.

Due to the spatial and vertical resolution of the RO technique, the waves analyzed here have relatively large wavelengths and
low frequencies. Despite this, the findings here should be important for TTL processes as Dzambo et al. (2019) shows that the
power spectrum of TTL gravity waves tend to peak at wavelengths of around 4–5 km, which is resolvable by RO soundings.
Nevertheless, it remains to be explored whether the Phase 1 of high-frequency waves are also distinct from other phases. Also,
possible explanations for Phase 1 favoring clouds remain an open question. Since negative $dT'/dz$ corresponds to a positive
cooling rate (due to downward phase propagation as explained by K16), weakened stability, as well as upward vertical motion
wave anomalies (according to the gravity wave polarization relationships), whether one has a stronger role in favoring clouds
needs to be better understood.

*Data availability.* The atmPrf radio occultation dataset can be obtained from the COSMIC Data Analysis and Archive Center (https://cdaac-
www.cosmic.ucar.edu/). The Aura MLS Level 2 $H_2O$ product is available from the Goddard Earth Sciences Data and Information Ser-
vices Center (https://disc.gsfc.nasa.gov), and the 2C-ICE CloudSat/CALIPSO product is available on the CloudSat Data Processing Center



(www.cloudsat.cira.colostate.edu). The CALIPSO Level 2 Cloud Profile and Cloud Layer products are hosted on the NASA Atmospheric Science Data Center (https://eosweb.larc.nasa.gov).

## Appendix A

**A1**

*Author contributions.* KC designed and performed the study with suggestions from TL. Both authors contributed to the writing of this article.

*Competing interests.* The authors declare no conflicts of interest.

*Acknowledgements.* This study was supported by the NASA Earth and Space Sciences Fellowship 80NSSC17K0384. We thank Sergey
Sokolovskiy and Zhen Zeng for comments regarding the usage of radio occultation data and William Read regarding the Aura MLS $H_2O$
product.





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



**Figure 9.** Composite of cloud frequency with respect to local minima (first row) or maxima (second row) of $T'$. The columns correspond to composites made from subsets of $T'$ extrema with magnitudes greater or equal to 0.5 K (left column), 1.0 K(middle), and 1.5 K (right). Dashed horizontal indicates the position of the local $T'$ extrema. The third row shows the background frequency (see text for explanation). The fourth row (j)-(l) and fifth rows (m)-(o) are the cloud frequencies anomalies associated with cold or warm anomalies, calculated as the difference between the cloud frequency composites and the background cloud frequency. Contours in the bottom two rows are at intervals of 1% (dashed negative), matching the filled color contours which show values at or above the 95% significance level.

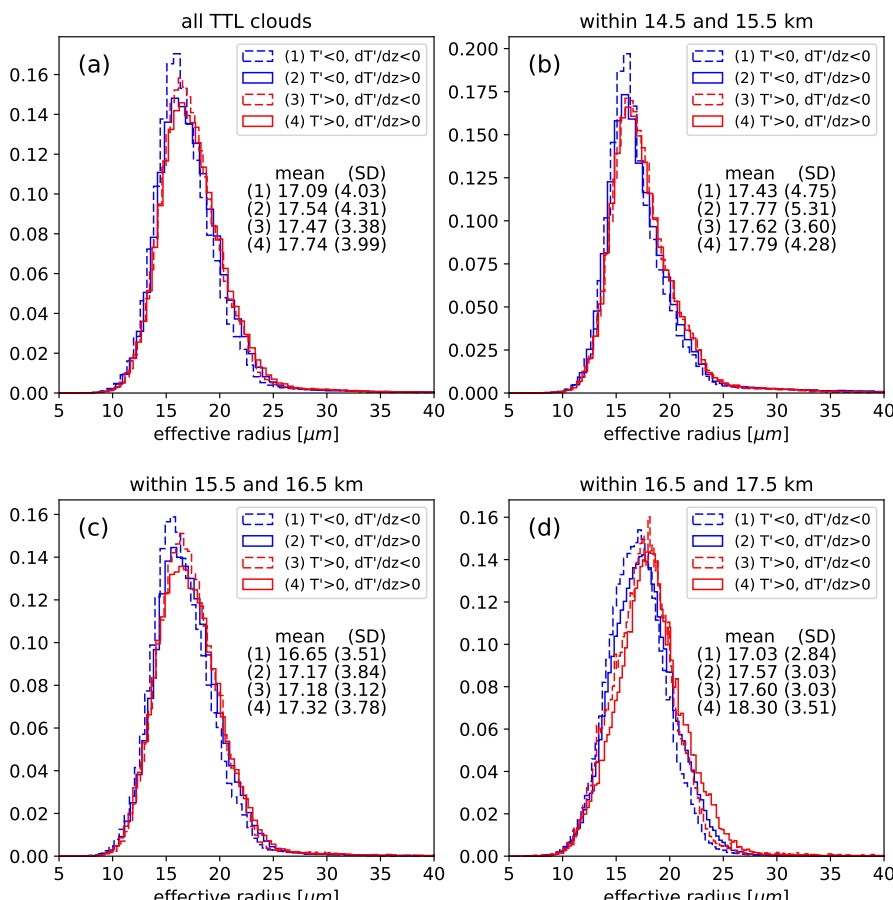

**Figure 10.** Normalized density function of $r_e$ in each gravity wave phase for (a) all of TTL and (b)-(d) 1-km vertical layers. The mean and standard deviation (SD) are given in the legend.





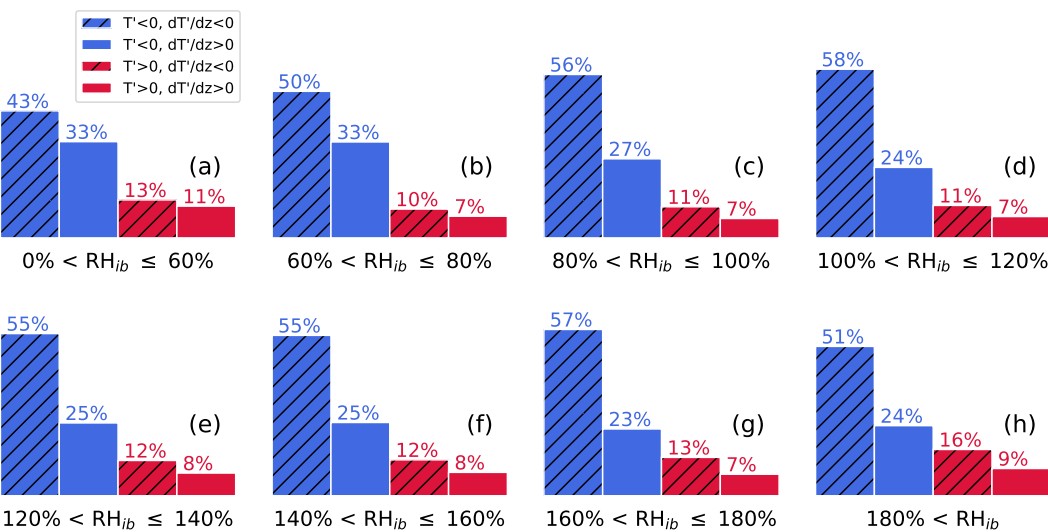

**Figure 11.** Percentage of TTL clouds inside each wave phase categorized by background relative humidity with respect to ice ($RH_{ib}$).