# Peer review of "Influence of gravity wave temperature anomalies and their vertical gradients on cirrus clouds in the tropical tropopause layer – a satellite-based view"

_Atmospheric Chemistry and Physics, 2020_

## Referee Comment (RC1) · Anonymous Referee #1 · 15 May 2020

Review of:

"Influence of gravity wave temperature anomaly and its vertical gradient on cirrus clouds in the tropical tropopause layer – a satellite-based view" by K-W Chang and T. L'Ecuyer

General Comments:

This paper extends the TTL gravity wave analyses of Kim et al. (2016), hereafter "K16" and Podglajen et al. (2018), hereafter "P18", by using satellite-derived temperature profiles using Radio Occultation (RO) techniques, cloud detection and effective ice crystal radius data from the Cloud-Aerosol Lidar and CloudSat radar from CALIPSO, and water vapor measurements from the Aura MLS. The focus is on identifying phases of TTL gravity waves most likely to generate and maintain ice clouds, and what environmental conditions influence their behavior.

I found the paper to be a measured expansion of previous work on the topic. I do think the authors need to perform some additional analyses to expand the scope and to focus and/or sharpen their findings, and provide additional explanations at certain points of the text, as outlined in my specific comments. Overall, I find the manuscript to be well-organized and written, so while some added study is desired, my suggested technical corrections are brief.

Specific Comments (by Line Number):

45: Your narrative suggests P18 agreed with K16 in placing the highest cloud frequencies in phase 1 of the gravity wave perturbation cycle. But in P18's figure 3, large ice crystals can also be found in phase 3 (where they are sublimating). Putting aside the effect of background relative humidity, P18's abstract says "The precise location where the confinement (i.e., wave-driven localization of ice crystals) occurs … is always characterized by… a positive vertical wind anomaly." So while P18 agrees that ice is suspended in phase 1, I would mention their finding that upward vertical motions permitted the presence of TTL ice clouds within phase 3 as well.

76: Why is the effective vertical resolution of RO profiles ~200m when temperature estimates are given at 30m spacing? Are the temperature data too noisy, and require vertical averaging? Are you using a version of the RO temperature retrievals that removes fine scale artifacts?

90: Explain what a "retrieval to flight ratio" means. Is that the ratio of the satellite-retrieved $r_e$ to that determined from aircraft measurements? How does the +/- 20% uncertainty of retrieved $r_e$ values compare with those derived from in-situ flight measurements?

140: How large is the sample of CALIPSO cloud profiles corresponding to each RO profile? In terms of the spatial collocation criteria I'd like to see the results of a sensitivity analysis, where you try both a smaller- and larger-sized diameter range with respect to the RO perigee point. Depending on sample size, I might try 50 km and 200 km and see if your results are affected in any significant way. A larger diameter incorporates more data, which is statistically preferable. But as wave phase surfaces slope with increasing distance from the RO perigee point, a smaller

diameter mitigates the possibility of associating the presence of clouds with the incorrect phase. Bound by these constraints, what spatial collocation criteria is optimal?

145: What is the mean (and standard deviation) of tropopause altitude in your RO dataset? It would follow that convective detrainment would only occur below this level; is this consistent with your finding that warm phase clouds decrease markedly above 16.5km? Or above an even lower altitude? Does this impact your comparison with K16's cloud fraction results? How so?

155: I think it would be worth re-running this analysis, having deleted altitudes that are below the mean tropopause for each geographical region and season. Does this reduce the influence of deep convection that is likely impacting your results for the western Pacific winter, or the Asian monsoon region during summer? Then either update figure 4 with new "filtered" results or else explain the difference (or lack thereof).

165: In regards to your comparison with K16's ATTREX results, I recommend re-computing your RO-derived T' values using a 30-day rather than a 7-day mean temperature. Does making this change result in a significant difference?

168: Re-word "line segments in Figure 1 according to each phase." Do you mean "…defined as the amount of vertical overlap between individual cloud boundaries and wave phases?"

For example, what if a profile's Phase 1 region extends from 14.8 – 15.2 km, but the cloud layer product indicates a cloud only from 14.9 – 15.1 km. In that layer, is CF = 0.2km/0.4km = 0.5?

Table 1: I wonder if this is necessary or can you just incorporate this information into the text? If you keep the table, I would (at least) eliminate the CF=0 column, and explain why the number of cases where CF=1 far exceeds the number of cases for which $0 < CF < 1$ (shown in figure 5).

207: Are the weak anomalies of figure 7 simply the result of variability in sampled cloud top heights diluting the overall result? Explain.

221: How can you compute cloud fractions with 50m vertical spacing when the CALIOP 5 km cloud profile product only reports the presence of cloud every 60m?

238-250: This explanation seems overly detailed and tedious. I would shorten this narrative, focusing on your statistical goals, and eliminate Figure 9(g)-(i), since these panels just look like averaged cloud fractions within the column, and show very minor differences between them. I would attempt to make the remaining 12 panels of figure 9 each a little bigger for visibility.

272-281: I'm surprised that distributions of $r_e$ from 2C-ICE data are so uniform. I doubt that the difference between altitude bins is significant. I'd prefer that in figure 10, you keep the results for all TTL altitudes (panel (a)), but delete the vertical stratification presented in panels (b)-(d). Instead, I'd like to see three panels of $r_e$ distributions from a more limited temporal and spatial domain (but for all altitudes combined), namely: (a) the region from 180W-120W during

DJF; (b) from 120E-180E during DJF; and (c) from 60E-120E during JJA. Then as per figure 4, discuss whether convective influence might have an impact on observed ice crystal sizes.

329: I would interpret this a bit differently. I think there is a clear trend in partitioning between phases 1 and 2 from low $RH_{ic}$ values, up to 100%. Thereafter it makes little difference, except for the highest supersaturation values.

Technical Corrections:

Title: Suggest "Influence of a gravity wave temperature anomaly…"

183: Try "-35, -33, … , -1, 1, … 33, 35"

Figure 4: Can you include the number of profiles that contain clouds for each latitude bin?

298-300: Try something like, "We conclude that our analyses of $r_e$ and $RH_{ic}$ data is qualitatively consistent with P18's results."

345-348: Suggest, "…as well as upward vertical motion all impact wave anomalies (…), it remains to be determined whether one has a stronger role in favoring cloud formation."

Figure 11: Add percentage symbols (%) to each of the $RH_{ic}$ labels and add the number of cloud encounters each panel represents.

---

## Author Comment (AC1) · 24 Jun 2020

To Referee 1: Thank you for reviewing this manuscript in detail. We appreciate the suggestions you gave, and think that they have led to considerable improvement of the manuscript. The comments are addressed below; green text is our response to each. Also, we made some general changes in the manuscript: 1. RHic was changed to RHib, and 2. the date rage of data used to produce the cloud populations has changed. The rationale is given below.

**General changes:**

- RHic (which was meant to stand for RHi climatology) has been changed to RHib (background RHi) for consistency with the terminology used in the discussion.
- In the submitted manuscript, we have used data within 1 Nov 2006 to 30 April 2014. We think that this causes an uneven sampling in season, which could affect the results of Figure 5. For this reason, in the revised version we plan to remake Figure 3-7 and Figure 9 using data within 1 Jan 2007 to 31 Dec 2013, so that the number of sampled months in each season is the same. After making this change, the largest change in percentage is 3% in Figure 3 and 1% in Figure 4. As an example, below is the new Figure 3. The resulting patterns in Figure 6, 7, and 9 are almost identical.

Figure 1. Same as Figure 3 in the manuscript, except made with data within 1 Jan 2007 to 31 Dec 2013.

**Specific Comments (by Line Number):**

45: Your narrative suggests P18 agreed with K16 in placing the highest cloud frequencies in phase 1 of the gravity wave perturbation cycle. But in P18's figure 3, large ice crystals can also be found in phase 3 (where they are sublimating). Putting aside the effect of background relative humidity, P18's abstract says "The precise location where the confinement (i.e., wave-driven localization of ice crystals) occurs ... is always characterized by... a positive vertical wind anomaly." So while P18 agrees that ice is suspended in phase 1, I would mention their finding that upward vertical motions permitted the presence of TTL ice clouds within phase 3

as well.

We have changed the wording of the sentence beginning with "P18 argues ...". We added a statement in the first paragraph of Section 4.1 that Phase 3's vertical motion permits the presence of ice.

76: Why is the effective vertical resolution of RO profiles ~200m when temperature estimates are given at 30m spacing? Are the temperature data too noisy, and require vertical averaging? Are you using a version of the RO temperature retrievals that removes fine scale artifacts?

It is probably more appropriate to say that the vertical 'weighting' is ~200m. At each height, the vertical weighting is more or less centered on the reported height in atmPrf, so each height has a slightly different vertical weighting even though they are only ~30m apart. In other words, the RO temperature can be interpreted as a temperature profile with a smoothing window applied, with the width of the window as the vertical weighting.

90: Explain what a "retrieval to flight ratio" means. Is that the ratio of the satelliteretrieved re to that determined from aircraft measurements? How does the +/- 20% uncertainty of retrieved re values compare with those derived from in-situ flight measurements?

Yes, the ratio is the satellite-retrieved re divided by the r\_e derived from in-situ 2D stereo probe measurements. The 1.05 is the mean ratio and we have revised the paper to indicate 'mean retrieval-to-flight ratio'. It is hard to know how our choice of +/-20% uncertainty compare with in-situ measurements. At line 280 we have noted that retrieval of re has large uncertainties especially with respect to thin cirrus., so that the reader is aware of the associated uncertainties in our results pertaining to r\_e.

140: How large is the sample of CALIPSO cloud profiles corresponding to each RO profile? In terms of the spatial collocation criteria I'd like to see the results of a sensitivity analysis, where you try both a smaller- and larger-sized diameter range with respect to the RO perigee point. Depending on sample size, I might try 50 km and 200 km and see if your results are affected in any significant way. A larger diameter incorporates more data, which is statistically preferable. But as wave phase surfaces slope with increasing distance from the RO perigee point, a smaller diameter mitigates the possibility of associating the presence of clouds with the incorrect phase. Bound by these constraints, what spatial collocation criteria is optimal? Figure 2 below shows the distribution for how many CALIPSO profiles tend to be collocated to be each RO profile. There is a sharp cutoff at around 40 profiles. Since each CALIPSO profile has 5-km horizontal resolution, 40\*5km is 200 km which is consistent with the 100-km radius collocation criterion.

As you say, a smaller diameter is probably more preferable since it is less likely to be influenced by sloping wave phases. We have done some testing using 50 km and 200 km as a collocation radius. As a reminder, the radius used in the manuscript is 100 km. Please see below for the cloud population in phases obtained using these collocation radii.

For altitudes below 17.5 km, the results from using 50 km vs 100 km (Figure 3 in this document vs Figure 1), the difference is at most 1%. Above 17.5 km, there are larger differences, but the number of samples is quite small at this altitude. Overall, there is good agreement between using 100 km and 50 km, which is reassuring. Results from using 200 km (Figure 4) remains qualitatively similar, but we see 2% differences (53% vs 55% in Phase 1 in panel (a)).

It is hard to objectively assess what collocation criterion is optimal. However, here we show that the results are not particularly sensitive to the collocation diameter. The qualitative features remain largely the same. Since 100 km provides more samples than 50 km, we prefer to maintain the use of 100 km. We will mention that we tested these collocation radiuses and the results had small variations in the text.

**Figure 2.** Distribution of the number of CALIPSO profiles collocated to RO profiles. The collocation criteria are 100-km radius and +-2 hours.

Figure 3. Same as Figure 1 except made with collocation radius of 50 km.

---

## Referee Comment (RC2) · Aurelien Podglajen (Referee) · 26 Jun 2020

In this paper, remote sensing satellite observations of cirrus clouds, temperature and water vapor are combined to analyze the relationship between ice clouds in the tropical tropopause layer and gravity waves. Using the much larger statistical sample enabled by satellite measurements, the authors confirm previous findings obtained from in situ observations. Furthermore, they expand on previous work by evaluating the qualitative consistency between observations and theoretical predictions linking ice cloud properties to different wave phases. In particular, they examine the influence of background

relative humidity on the gravity wave-cirrus relation and the ice crystal size in different wave phases.

Overall, I find this study thorough and clearly written. The authors designed enlightening composite diagnostics and present rigorous significance tests of their results. The article is definitely worth publication in ACP. I only have a number of minor comments and suggestions for the authors' consideration, which are detailed below. Since my review comes at a late stage in the ACP discussion process (apologies) and since the authors have already answered referee 1, my comments concern the manuscript as well as the reply to reviewer 1. I mostly agree with the other reviewer's suggestions but have spotted a misunderstanding regarding the P18 paper.

**Minor comment**:

Contrary to Kim et al (2016), the authors show the distribution of clouds between the different wave phases instead of the mean cloud fraction in each phase. The former can be deduced from the latter (which the authors do on page 6 line 148), while the reverse is not true. I am a bit puzzled by this choice which appears to result in a reduction of the information conveyed by the figures.

Is it motivated by the sensitivity of the mean cloud fraction to the CAD threshold used to distinguish clouds from aerosols? By the fact that the cloud fraction depends on CALIOP optical depth detection threshold? Or is it just meant to facilitate the visual and quantitative comparison between regions/altitudes with different cloud fractions? Some explanation would be welcome.

**Specific comments**

- page 2, line 44-46: A clarification here related to Referee 1's comment: "while P18 agrees that ice is suspended in phase 1, I would mention their finding that upward vertical motions permitted the presence of TTL ice clouds within phase 3 as well" and the authors reply. Actually, the interpretation in the submitted

manuscript was correct: P18 indeed claim that phase 1 is more favorable for cirrus than phase 3. Referee 1's confusion lies in the choice of words in the quoted sentence. The exact P18 sentence reads: "The precise location where the confinement occurs ... is always characterized by a relative humidity near saturation and a positive vertical wind anomaly" . The effect of relative humidity cannot be put aside to correctly interpret that sentence. Indeed, relative humidity near or above saturation is mostly found in negative temperature anomalies (Phase 1 or 2) while the vertical wind is positive in phases 1 and 3. Taken together, the two conditions mean that clouds are found in phase 1. This is illustrated by figure 2 of P18. Note that the sketch in their figure 3 was meant for purely pedagogic purposes, showing possible cases.

- P3 line 83-85 : Are the results affected by the choice of the CAD threshold ? By how much does the average TTL cloud fraction estimate vary depending on this threshold?

- P3 lines 88-92 and fig. 10 : I believe that in most thin TTL cirrus there are only Lidar measurements available (their reflectivity is below the radar detection threshold), so that re cannot be unequivocally determined. This might explain why the 2D ice data are so uniform.

- P5 line 128-130: This is more of a sanity check, but are all 4 phases equiprobable ?

- P6 line 147: you should specify that K16 measurements were made in the Western Pacific

- p7 lines 160: The contrast remains, but I would say that K16 found "slightly" more clouds in phase 2 (6 % vs 5 % in phase 1).

- p7 line 172 and Figure 5: Instead of the count (number of layers?), would it be

clearer to show some "phase fraction" (i.e. normalized by cumulated altitude)? I am unsure myself.

- p8 line 177-182: I like the composite approach, but it might be worth mentioning that, by essence, it is stationary in space, so that neither the cloud (moved by the wind) nor the wave anomaly (moving at the wave phase speed) are followed. For instance, if the structure moves at 10 m/s, 30 hours corresponds to a displacement more than 1,000 km (10 times the collocation radius) away from the location where the composite is made.

- P10 lines 202-203: I don't think the roles of the wave through its impact on stability and onvertical motion can be easily disentangled. They necessarily hold at the same time, as a consequence of the polarization relations.

- P10 lines 205-209: I am wondering whether the typical vertical wavelength of 3km which is emphasized by this composite means something.

- p13 line 266: The confinement hypothesized in P18 can only occur in Phase 1 (see their Fig 2.), not in Phase 2

- p13 line 270: Note that P18 derivation is also valid for a constant background wind (the exact same set of equations are obtained considering a frame moving at the constant background wind speed). However, it is true that the background wind shear cannot be as easily included in their approach and is neglected.

- P13 line 280 : A prediction of P18 is that the fall speed is comparable to the GW vertical phase speed. To get an idea, could you compare the fall speed of a crystal (assumed spherical for simplicity) of that radius to to the typical GW vertical phase speed in Fig. 6?

- P15 line 285: Again, P 18-type confinement always occurs in phase 1 for a monochromatic wave. With lower Rhic, the location gets closer to the boundary between phase 1 and 2, so that with a superposition of waves both phases might show similar cloud fraction.

- P 15 lines 290-300: It is an interesting approach that the authors attempt here, and I like the new Fig 11 in the reply to referee 1. However, I see small issues with the method employed by the authors.

  First, the non-linearity of the Goff-Gratch equation means that the coarse-resolution MLS water vapor divided by the saturation pressure of the mean temperature will lead to a larger relative humidity than the average relative humidity (average ratio of the two). This might explain why the authors estimates seem slightly high-biased compared with in situ observations of TTL humidity (see for instance Jensen et al., 2017 for a survey of TTL relative humidity from in situ measurements). This might result in a systematic bias in the authors' estimate. However, I imagine that the trend found by the authors will not be not sensitive to this.

  Second, as far as I understand, the temperature is estimated from 7-day averages but the water vapor from instantaneous values. For consistency, the water vapor should be taken from averages as well.

- p16 line 343: 4-5 km seems larger than the typical wavelength which comes out of your composites Fig. 6

- p16 lines 345-348: "Since negative $dT_0/dz$ corresponds to a positive cooling rate (due to downward phase propagation as explained by K16), weakened stability, as well as upward vertical motion wave anomalies (according to the gravity wave polarization relationships),": the two points " positive cooling rate" and "upward vertical motion" are equivalent under the usual adiabatic approximation. One should be removed from the sentence.

**Wording**:

- P1, line 20 :Maybe replace "favor" by "is favorable to" or use a passive.

- p7 line 167: 'vertical cloud fraction': I would remove vertical.

- P7 line 168: 'cloud boundaries' → 'cloud layer' ?

- P7 line 175: Again, I would put a passive form.

- P10 line 213 : "a similar compositing technique ..."→ "a compositing technique similar to the one employed above "

- P16 l340 : large wavelengths → large horizontal wavelengths

**Reference**

Jensen, E. J., et al. (2017), Physical processes controlling the spatial distributions of relative humidity in the tropical tropopause layer over the Pacific, J. Geophys. Res. Atmos., 122, 6094– 6107, doi:10.1002/2017JD026632.

---

## Referee Comment (RC3) · Anonymous Referee #1 · 30 Jun 2020

Overall: I appreciate you incorporating my suggestions. They have improved the original manuscript in a significant way, and I fully expect your paper will be published after you complete the changes you've indicated are in process. Below I offer just a few additional thoughts on specific matters.

Specific Comments (by Line Number):

45: I need to clarify my previous comment to conform to the definition of the four gravity wave phases that you provided at the top of section 4 (lines 128-130 in the original MS).

If I understand correctly, P18's figure 3 appears to show the possibility of ice crystals present in three of the four phases. These would be phase 1 (air is cooling due to upward vertical motion, leading to ice nucleation), phase 2 (air begins to warm due to downward vertical motion, but in the presence of ice supersaturation, ice crystals may grow larger by vapor deposition), and possibly, phase 4, where sublimation occurs along with downward motion, shrinking any remaining crystals and returning their water content to the vapor state.

In my review, my mistake was to confuse phase 3 with phase 4. As the other reviewer, Aurelian has weighed in to say that his figure 3 is purely "pedagogic," i.e., that he didn't mean to imply that clouds could exist where RHi < 100% (phase 3 or 4), but I slightly disagree, since once ice crystals have grown to reach a maximum size, they can still exist for a finite time in downward moving, sub-saturated air. Granted, whether they would do so would highly depend on individual wave morphology as well as the localized temperature and moisture profiles.

Figure 4 update: It does really seem as if clouds formed by waves (i.e., above the tropopause) are less likely to be observed in both warm phases (regardless of season), but especially so in phase 3. Clouds seem to be a bit more likely to be detected in phase 2 during DJF.

Figure 11 update: I like the new line plot showing the dependence of cloud presence on RHib within phases 1 and 2, especially in sub-saturated conditions, as well as the distribution of RHib in clouds.

---

## Referee Comment (RC4) · Anonymous Referee #1 · 30 Jun 2020

Hi Aurelien,

I'll briefly respond to your clarification relating to my comment regarding lines 44-46 of the original MS. As I've indicated to the authors in my response to their list of corrections, I was confused about the numbering of the four wave phases. I meant to refer to phase 4 instead of phase 3. Also, as I didn't really intend to put aside the effect of ice saturation, let me explain how I understand relative humidity variations to affect the process of air moving through the wave.

[Figure]

Phase 3 should be in fact, the least likely quadrant to encounter ice crystals. As I explained to the authors, air initially rising & cooling during phase 1 will (given sufficient supersaturation) nucleate ice crystals. During phase 2, the air is starting to sink and warm, but is still cold with RHi near or above 100%, allowing crystals that have already formed to grow larger by vapor deposition. What would be observed next in phase 4, where the warming, sinking air becomes sub-saturated, would really depend on how large the ice crystals grew, the particular wave motions, and the background state conditions. It seems that if available moisture is lacking, the crystals would be pretty much disappear while in phase 2, leaving phase 4 cloudless. However, if there's a lot of available vapor, some crystals could survive for a short time as the wave processes an air parcel through phase 4 ($w' < 0$ with $dT'/dZ > 0$).

I think your "P18" figure 3, while "pedagogic" is still valid in principle. The authors of the current manuscript will be changing their figure 4 to compare cloud distributions observed above the tropopause with those found within the TTL in general. Granted, the number of samples confined to the upper TTL is somewhat sparse compared to the larger dataset. Still, it would appear that filtering out the convective influence from the upper troposphere seems to shift the cloud distribution to increase the proportion of clouds in phase 2, and also has a slightly greater impact on reducing cloud encounters within phase 3 than in phase 4. Granted, the signal in the warm phases may be rather faint in these CALIPSO cloud observations.

I appreciate your comment and hope this clarifies my position.

Reviewer #1
* * *

---

## Author Comment (AC2) · 20 Jul 2020

Hello Referee 1,

In our previous reply regarding your comment to Figure 10, we said that we will use the new Figure 10 ($r\_e$ in different longitude belts). However, we think that the results in this new figure is rather inconclusive and doesn't show anything significant. For this reason, we are planning to submit the revised manuscript using the original Figure 10 ($r\_e$ stratified by height). I hope this is acceptable. We made minor modification to this

Figure by adding the total number of r_e samples in each plot.

Best, Kai-Wei

———————————————————

---

## Author Response (AR1)

This is an updated version of our response to Referee 1. Our response and changes to the manuscript are indicated separately, and Referee 2's comments have also been considered in the responses below. Referee comments are in black, author responses are in green, and manuscript changes are in blue.

**General changes:**

We made some manuscript changes that are not based on the referee's comments. These are:

- RHic (which was meant to stand for RHi climatology) has been changed to RHib (background RHi) for consistency with the terminology used in the discussion.

- In the submitted manuscript, we have used data within 1 Nov 2006 to 30 April 2014. We think that this causes an uneven sampling in season, which could affect the results of Figure 5. For this reason, in the revised version we plan to remake Figure 3-7 and Figure 9 using data within 1 Jan 2007 to 31 Dec 2013, so that the number of sampled months in each season is the same. After making this change, the largest change in percentage is 3% in Figure 3 and 1% in Figure 4.

**Specific Comments (by Line Number):**

45: Your narrative suggests P18 agreed with K16 in placing the highest cloud frequencies in phase 1 of the gravity wave perturbation cycle. But in P18's figure 3, large ice crystals can also be found in phase 3 (where they are sublimating). Putting aside the effect of background relative humidity, P18's abstract says "The precise location where the confinement (i.e., wave-driven localization of ice crystals) occurs … is always characterized by… a positive vertical wind anomaly." So while P18 agrees that ice is suspended in phase 1, I would mention their finding that upward vertical motions permitted the presence of TTL ice clouds within phase 3 as well.

Author Response: Based on Referee 2's response to this point, we have decided to leave the sentence as-is.

Manuscript changes: At line 190 we mention that in Phase 3 the upward vertical wind anomaly may play a role in it having more clouds than Phase 4.

76: Why is the effective vertical resolution of RO profiles ~200m when temperature estimates are given at 30m spacing? Are the temperature data too noisy, and require vertical averaging? Are you using a version of the RO temperature retrievals that removes fine scale artifacts?

Author Response: It is probably more appropriate to say that the vertical 'weighting' is ~200m. At each height, the vertical weighting is more or less centered on the reported height in atmPrf, so each height has a slightly different vertical weighting even though they are only ~30m apart. In other words, the RO temperature can be interpreted as a temperature profile with a

smoothing window applied, with the width of the window as the vertical weighting.
Manuscript Changes: No changes.

90: Explain what a "retrieval to flight ratio" means. Is that the ratio of the satellite-retrieved re to that determined from aircraft measurements? How does the +/- 20% uncertainty of retrieved re values compare with those derived from in-situ flight measurements?

Author Response: Yes, the ratio is the satellite-retrieved re divided by the $r_e$ derived from in-situ 2D stereo probe measurements. The 1.05 is the mean ratio and we have revised the paper to indicate 'mean retrieval-to-flight ratio'. It is hard to know how our choice of +/- 20% uncertainty compare with in-situ measurements. At line 280 we have noted that retrieval of re has large uncertainties especially with respect to thin cirrus., so that the reader is aware of the associated uncertainties in our results pertaining to $r_e$.

Manuscript Changes: No changes.

140: How large is the sample of CALIPSO cloud profiles corresponding to each RO profile? In terms of the spatial collocation criteria I'd like to see the results of a sensitivity analysis, where you try both a smaller- and larger-sized diameter range with respect to the RO perigee point. Depending on sample size, I might try 50 km and 200 km and see if your results are affected in any significant way. A larger diameter incorporates more data, which is statistically preferable. But as wave phase surfaces slope with increasing distance from the RO perigee point, a smaller diameter mitigates the possibility of associating the presence of clouds with the incorrect phase. Bound by these constraints, what spatial collocation criteria is optimal?

Author Response:

Figure 2 below shows the distribution for how many CALIPSO profiles tend to be collocated to be each RO profile. There is a sharp cutoff at around 40 profiles. Since each CALIPSO profile has 5-km horizontal resolution, 40*5km is 200 km which is consistent with the 100-km radius collocation criterion.

As you say, a smaller diameter is probably more preferable since it is less likely to be influenced by sloping wave phases. We have done some testing using 50 km and 200 km as a collocation radius. As a reminder, the radius used in the manuscript is 100 km. Please see below for the cloud population in phases obtained using these collocation radii.

For altitudes below 17.5 km, the results from using 50 km vs 100 km (Figure 3 in this document vs Figure 1), the difference is at most 1%. Above 17.5 km, there are larger differences, but the number of samples is quite small at this altitude. Overall, there is good agreement between using 100 km and 50 km, which is reassuring. Results from using 200 km (Figure 4) remains qualitatively similar, but we see 2% differences (53% vs 55% in Phase 1 in panel (a)).

It is hard to objectively assess what collocation criterion is optimal. However, here we show that the results are not particularly sensitive to the collocation diameter. The qualitative features remain largely the same. Since 100 km provides more samples than 50 km, we prefer to maintain the use of 100 km. We will mention that we tested these collocation radiuses and the results had small variations in the text.

[Figure]

**Figure 2.** Distribution of the number of CALIPSO profiles collocated to RO profiles. The collocation criteria are 100-km radius and +-2 hours.

[Figure]

**Figure 3.** Same as Figure 1 except made with collocation radius of 50 km.

[Figure]

**Figure 4.** Same as Figure 1 except made with collocation radius of 200 km.

Manuscript Changes: No changes.

145:    What is the mean (and standard deviation) of tropopause altitude in your RO dataset?    It would follow that convective detrainment would only occur below this level; is this consistent with your finding that warm phase clouds decrease markedly above 16.5km? Or above an even lower altitude?    Does this impact your comparison with K16's cloud fraction results?    How so?

Author Response:

The mean and standard deviation of the tropopause within 20N/S of the equator is 17.14 and 0.72 km. For a detailed discussion on the differences of clouds above or below the tropopause, please see our response to the next comment. In summary, below the tropopause we find more clouds in the warm phase, consistent with the expectation that convective detrainment occurs below the tropopause level.

In K16's Figure 5, they showed that the T' of low clouds (below 16 km) associated with convection were distributed evenly across the four phases, while clouds non associated with convection were more common in Phase 1 and 2. Although we are not able to separate convective and non-convective clouds, our results here are consistent with their findings.

Manuscript Changes: No changes

155:    I think it would be worth re-running this analysis, having deleted altitudes that are below the mean tropopause for each geographical region and season. Does this reduce the influence of deep convection that is likely impacting your results for the western Pacific winter, or the Asian monsoon region during summer?    Then either update figure 4 with new "filtered" results or else explain the difference (or lack thereof).

Author Response:

We have updated Figure 5 to include the cloud population above the tropopause. The

mean tropopause for each longitude band and season was calculated using RO. There are generally less clouds in the warm phases and more in the cold phases when looking only above the tropopause. In the western Pacific winter region, there is a significant reduction of clouds in the warm phase. The reduction is less apparent in the summer Asian monsoon. One possibility is that the tropopause is elevated by the upper tropospheric anticyclone (associated with the monsoon), causing outflow clouds to be present at higher altitudes. In Section 4.1 we will add discussion regarding the exclusion of clouds below the tropopause. Manuscript Changes: Changed Figure 4, and also changed the discussion in Section 4.1 regarding this figure.

165: In regards to your comparison with K16's ATTREX results, I recommend re-computing your RO-derived T' values using a 30-day rather than a 7-day mean temperature. Does making this change result in a significant difference?
Please see Figure 6 below for the cloud population derived using 31-day mean temperature. Compared to the population derived from 7-day means, there tends to be slightly more clouds in the Phase 1 and less clouds in the warm phases. Unlike what K16 found, we do not see that Phase 2 having more clouds than Phase 1 in the 180W-120W band. In paragraph 4 of Section 4.1 we will add a note that we tried using the 31-day mean and explain what we found.

[Figure]

**Figure 6**: Same as Figure 4 in the manuscript, except derived using 31-day mean temperature as background state.

Manuscript Changes: Added discussion regarding the use of 7-day mean vs 31-day mean at the end of the 4ᵗʰ paragraph in Section 4.1 (around line 175)

168: Re-word "line segments in Figure 1 according to each phase." Do you mean "…defined as the amount of vertical overlap between individual cloud boundaries and wave phases?" For example, what if a profile's Phase 1 region extends from 14.8 – 15.2 km, but the cloud layer product indicates a cloud only from 14.9 – 15.1 km. In that layer, is CF = 0.2km/0.4km = 0.5?

Author Response: Yes, we mean that it is the amount of overlap between individual cloud boundaries and wave phases, as you stated. We have edited the sentence to state this. Your example is correct, in that case the CF is 0.5.

Manuscript Changes: Added an example like the one here at line 179 of the revised manuscript.

Table 1: I wonder if this is necessary or can you just incorporate this information into the text? If you keep the table, I would (at least) eliminate the CF=0 column, and explain why the number of cases where CF=1 far exceeds the number of cases for which 0 < CF < 1 (shown in figure 5).

Author Response:

For Table 1, we have decided to put the total count of 0<CF<1 in place of the C=0 column, and in the CF=1 column we have added a percentage showing the fraction of instances with CF=1.

As for why CF=1 tends to outnumber cases with 0<CF<1, see Figure 7. We find that the vertical thicknesses of the Phase 1 (distribution in blue) tend to be smaller than the typical cloud thickness (black). Since clouds are thicker, it is likely that the entire Phase 1 is embedded in clouds, hence the higher frequency.

[Figure]

**Figure 7**: Distribution of Phase 1 vertical thickness (of those used to derive Figure 5 of the manuscript), and the thicknesses of clouds that overlap with Phase 1 (black)

Manuscript Changes: Modified Table 1 and discussion regarding this table.

207:    Are the weak anomalies of figure 7 simply the result of variability in sampled cloud top heights diluting the overall result? Explain.

Author Response: Yes, the weak anomalies are a result of cloud top height variability. Since it is common that TTL clouds are embedded in Phase 1, their cloud tops tend to be close to the T' minimum (this is depicted in Figure 6). Since we don't composite with respect to the cloud top height in Figure 7, the result is a pattern of weak anomalies.

Manuscript Changes: No changes.

221:    How can you compute cloud fractions with 50m vertical spacing when the CALIOP 5 km cloud profile product only reports the presence of cloud every 60m?

Author Response: When the CALIPSO product reports the presence of a cloud in a bin, we are assuming that the cloud occupies the entirety of the bin. Say that the CALIPSO product reports that a bin at height z has a cloud. Then we assume that the region between $z0 = z + 30m$ and $z1 = z - 30m$ is occupied by a cloud. Then, in the grid on which we calculate the cloud fraction (the grid has 50-m spacing), we check the overlap of the cloud boundary $[z0,z1]$ with each bin on the grid. In this approach, the two spacings do not need to match. Because the reported CALIPSO heights are not on a regular height grid, there isn't much advantage in using a 60-m grid since the same method would have to be used.

Manuscript Changes: No changes.

238-250:    This explanation seems overly detailed and tedious. I would shorten this narrative, focusing on your statistical goals, and eliminate Figure 9(g)-(i), since these panels just look like averaged cloud fractions within the column, and show very minor differences between them. I would attempt to make the remaining 12 panels of figure 9 each a little bigger for visibility.

Manuscript Changes: We have moved the explanation (most of lines 238-250 of the submitted manuscript) to the appendix, and in the main text we briefly summarize that we extract the anomalies by building a background cloud fraction. Figure 9(g)-(i) has been eliminated from Figure 9.

272-281:    I'm surprised that distributions of re from 2C-ICE data are so uniform.   I doubt that the difference between altitude bins is significant. I'd prefer that in figure 10, you keep the results for all TTL altitudes (panel (a)), but delete the vertical stratification presented in panels (b)-(d).   Instead, I'd like to see three panels of re distributions from a more limited temporal and spatial domain (but for all altitudes combined), namely: (a) the region from 180W-120W during DJF; (b) from 120E-180E during DJF; and (c) from 60E-120E during JJA. Then as per figure 4, discuss whether convective influence might have an impact on observed

ice crystal sizes.

Author Response:

We have reproduced the r_e distribution as you specified (shown below) The regions associated with frequent convection (120E-180E DJF and 60E-120E JJA) exhibit a lower mean r_e in Phase 1 and 3. The r_e distribution of these phases also tend have a higher peak compared to Phase 2 or 4. However, it is not clear whether this is a result of convection or not. One might have expected to see larger r_e in these regions due to detrainment, yet this is not observed. An interesting feature is that the distributions over 60E-120E JJA are notably narrower than those of 120E-180E DJF, but this feature seems independent of the wave phase.

In the previous response to Referee 1, we wrote that we will use this new figure in the paper. However, the new figure does not suggest any significant influence from convection, so the results are inconclusive. Considering this, we decided to keep the original Figure and discussion.

Manuscript Changes: Added number of r_e samples to Figure 10, and updated the bin widths (all distributions now have the same bin widths).

[Figure]

329: I would interpret this a bit differently. I think there is a clear trend in partitioning between phases 1 and 2 from low RHic values, up to 100%. Thereafter it makes little difference, except for the highest supersaturation values.

Author Response This is an interesting point. Figure 11 has been remade into a line plot to better show the cloud population as a function of RHib. RHib is now binned in 0%, 50%, 60%, 70%, …, 180%. The 0% to 50% is grouped together because of the low number of samples. As shown below (Figure 9), there is a rather apparent trend in Phase 1 from 50% to 100% as you noted. On the other hand, Phase 3 and 4 don't seem sensitive to RHib. Phase 2 also exhibits a decreasing trend as RHib increases up to 100%. In the text, the discussion has been edited for this new figure.

Manuscript Changes: Updated Figure 11 and subsequent discussion.

**Technical Corrections:**

Title:    Suggest "Influence of a gravity wave temperature anomaly…"

Manuscript Changes: Title was changed to "Influence of gravity wave anomalies and their vertical gradients on cirrus clouds in the tropical tropopause layer -- a satellite-based view"

183:    Try "-35, -33, … , -1, 1, … 33, 35"

Manuscript Changes: Changed as suggested.

Figure 4: Can you include the number of profiles that contain clouds for each latitude bin?

Manuscript Changes: We have added the number of CALIPSO Cloud Profile bins included in each plot. The number is shown under the longitude label. Note that this isn't the same as the number of profiles since one profile may have multiple bins with clouds.

298-300:    Try something like, "We conclude that our analyses of re and RHic data is qualitatively consistent with P18's results."

Manuscript Changes: Changed as suggested.

345-348:    Suggest, "…as well as upward vertical motion all impact wave anomalies (…), it remains to be determined whether one has a stronger role in favoring cloud formation."

Manuscript Changes: Changed as suggested.

Figure 11:    Add percentage symbols (%) to each of the RHic labels and add the number of cloud encounters each panel represents.

Manuscript Changes: This figure has been changed from bar plots to line plots showing cloud population as a function of RHib (see Figure 9 above). The number of cloud bins in each RHib category is also shown in the bottom plot.

**Response to Referee 2's comments**

Thank you for the feedback on the submitted manuscript and Referee 1's comments. We have adopted most of your suggestions, as detailed below. Referee comments are in black, author responses are in green, and manuscript changes are in blue.

**Minor comment:**

Contrary to Kim et al (2016), the authors show the distribution of clouds between the different wave phases instead of the mean cloud fraction in each phase. The former can be deduced from the latter (which the authors do on page 6 line 148), while the reverse is not true. I am a bit puzzled by this choice which appears to result in a reduction of the information conveyed by the figures.

Is it motivated by the sensitivity of the mean cloud fraction to the CAD threshold used to distinguish clouds from aerosols? By the fact that the cloud fraction depends on CALIOP optical depth detection threshold? Or is it just meant to facilitate the visual and quantitative comparison between regions/altitudes with different cloud fractions? Some explanation would be welcome.

Author Response: With aircraft observations as in Kim et al. (2016), they can obtain the cloud fraction in each phase, presumably by dividing the number of observations with clouds by the total number of obs. With CALIPSO observations, the 'total' number of observations is somewhat arbitrary. For example, if we define the total number of obs to be all CALIPSO bins within 14.5 to 20 km, then the cloud fraction in all phases will be quite low, since above ~17 km most CALIPSO bins will have no clouds. As you can see the 'cloud fraction' derived this way can vary a lot depending on the TTL height bounds, so we decided to look at the partitioning among phases, which is straightforward to interpret.
Manuscript Changes: No changes made for this comment.

**Specific comments**

Page 2, line 44-46: A clarification here related to Referee 1's comment: "while P18 agrees that ice is suspended in phase 1, I would mention their finding that upward vertical motions permitted the presence of TTL ice clouds within phase 3 as well" and the authors reply. Actually, the interpretation in the submitted manuscript was correct: P18 indeed claim that phase 1 is more favorable for cirrus than phase 3. Referee 1's confusion lies in the choice of words in the quoted sentence. The exact P18 sentence reads: "The precise location where the confinement occurs ... is always characterized by a relative humidity near saturation and a positive vertical wind anomaly" . The effect of relative humidity cannot be put aside to correctly interpret that sentence. Indeed, relative humidity near or above saturation is mostly found in negative temperature anomalies (Phase 1 or 2) while the vertical wind is positive in phases 1 and 3. Taken together, the two conditions mean that clouds are found in phase 1. This is illustrated by figure 2 of P18. Note that the sketch in their figure 3 was meant for purely pedagogic purposes, showing possible cases.
Author Response: We will leave this sentence as-is, since the interpretation is consistent with P18

Manuscript Changes: No changes made for this comment.

P3 line 83-85 : Are the results affected by the choice of the CAD threshold ? By how much does the average TTL cloud fraction estimate vary depending on this threshold?

Author Response: The choice of CAD fraction doesn't affect the cloud fraction much. In terms of the percentage of all TTL clouds in each phase, a threshold of CAD >= 80 results in 53.4%, 25.9, 12.2, and 8.5 for Phases 1,4, while CAD>=90 yields 53.4%, 26.1%, 12.1%, and 8.5.

Manuscript Changes: No changes made for this comment.

P3 lines 88-92 and fig. 10 : I believe that in most thin TTL cirrus there are only Lidar measurements available (their reflectivity is below the radar detection threshold), so that re cannot be unequivocally determined. This might explain why the 2D ice data are so uniform

Author Response: Yes, it is true that radars miss most of the thin TTL cirrus. Aside from this, their being very optically thin also poses challenges in constraining the retrieval.

Manuscript Changes: No changes made for this comment.

P5 line 128-130: This is more of a sanity check, but are all 4 phases equiprobable?

Author Response: Yes, we've done some checks and found that the number of samples across the four phases are more or less evenly distributed.

Manuscript Changes: No changes made for this comment.

P6 line 147: you should specify that K16 measurements were made in the Western Pacific

Manuscript Changes: Added the clarification starting at line 152 of the revised manuscript.

P7 lines 160: The contrast remains, but I would say that K16 found "slightly" more clouds in phase 2 (6 % vs 5 % in phase 1).

Manuscript Changes: Added "slightly" in the at line 170 of the revised manuscript.

P7 line 172 and Figure 5: Instead of the count (number of layers?), would it be clearer to show some "phase fraction" (i.e. normalized by cumulated altitude)? I am unsure myself.

Author Response: You are correct that the count is the number of layers. We added an example of how the cloud fraction is calculated, in hopes of making this part clearer. We don't' quite understand what is meant by showing phase fraction normalized by cumulated altitude, and decided to leave the figure as-is.

Manuscript Changes: Added an example of the cloud fraction calculation at line 180 of the revised manuscript.

P8 line 177-182: I like the composite approach, but it might be worth mentioning that, by essence, it is stationary in space, so that neither the cloud (moved by the wind) nor the wave anomaly

(moving at the wave phase speed) are followed. For instance, if the structure moves at 10 m/s, 30 hours corresponds to a displacement more than 1,000 km (10 times the collocation radius) away from the location where the composite is made

Manuscript Changes: We added a sentence in the conclusion (line 351) to note the stationary nature of this technique.

P10 lines 202-203: I don't think the roles of the wave through its impact on stability and on vertical motion can be easily disentangled. They necessarily hold at the same time, as a consequence of the polarization relations.

Manuscript Changes: We modified this sentence here to be : "Since negative $dT'/dz$ also corresponds to upward vertical motion anomalies (assuming that these anomalies are from gravity waves), it is difficult to separate the effects of each on cloud formation."

P10 lines 205-209: I am wondering whether the typical vertical wavelength of 3km which is emphasized by this composite means something.

Author Response: We are not quite sure. One thing comes to mind, which is that Alexander et al. (200) found that temperature fields at dominated by waves of vertical wavelengths ~2 km, which is not so different than 3 km. However their result is for the tropical lower stratosphere 18 – 25 km so their findings may not have a direct connection to the ~3 km structure we found.

Manuscript Changes: No changes made.

P13 line 266: The confinement hypothesized in P18 can only occur in Phase 1 (see their Fig 2.), not in Phase 2

Author Response: We mentioned that ice can be found in Phase 2 based on P18's Figure 2 right panel, where for RHic=0.63 you can see a small portion of the ice within Phase 2. We reworded this sentence to clarify this.

Manuscript Changes: At line 271 of the new manuscript, the sentenence is now "P18 suggests that (1) ice crystals within a confined range of $r\_e$ are suspended in Phase 1, and (2) for low background relative humidity with respect to ice (RH_ib}), the confinement in Phase 1 may be positioned closer to Phase 2."

P13 line 270: Note that P18 derivation is also valid for a constant background wind (the exact same set of equations are obtained considering a frame moving at the constant background wind speed). However, it is true that the background wind shear cannot be as easily included in their approach and is neglected.

Manuscript changes: This sentence (now at line 274 of the revised manuscript) is changed to note that only wind shear is neglected in P18.

P13 line 280 : A prediction of P18 is that the fall speed is comparable to the GW vertical phase

speed. To get an idea, could you compare the fall speed of a crystal (assumed spherical for simplicity) of that radius to to the typical GW vertical phase speed in Fig. 6?

Author Response: Assuming a radius of 15 um and temperature of 200 K, P18's equation (20) (valid for radii of 5-100 um) yields a sedimentation velocity of ~ 2cm/s. This corresponds to a displacement of ~1.3 km over 18 hours. In Figure 6, using the cold anomaly in -12 to +6 hours, the descent rate is about 1 km, which is comparable to the ice sedimentation velocity.

Manuscript Changes: A paragraph is added at line 221 to describe P18's prediction regarding the fall speed, and how the descent rate of the cold anomaly rates to the estimated ice fall speed.

P15 line 285: Again, P18-type confinement always occurs in phase 1 for a monochromatic wave. With lower Rhic, the location gets closer to the boundary between phase 1 and 2, so that with a superposition of waves both phases might show similar cloud fraction.

Manuscript Changes: Sentence at line 290 is modified to say that in low RHic ice gets closer to Phase 2 (instead of being inside Phase 2).

P 15 lines 290-300: It is an interesting approach that the authors attempt here, and I like the new Fig 11 in the reply to referee 1. However, I see small issues with the method employed by the authors.

First, the non-linearity of the Goff-Gratch equation means that the coarse resolution MLS water vapor divided by the saturation pressure of the mean temperature will lead to a larger relative humidity than the average relative humidity (average ratio of the two). This might explain why the authors estimates seem slightly high-biased compared with in situ observations of TTL humidity (see for instance Jensen et al., 2017 for a survey of TTL relative humidity from in situ measurements). This might result in a systematic bias in the authors' estimate. However, I imagine that the trend found by the authors will not be not sensitive to this.

Second, as far as I understand, the temperature is estimated from 7-day averages but the water vapor from instantaneous values. For consistency, the water vapor should be taken from averages as well.

Author Response:

We will note in the text that the coarse resolution of the MLS and the nature of the Goff-Gratch equation may have contributed to the high-biased RH estimates. Regarding your second point, we have tried to calculate Figure 11 with 7-day averages of moisture, as shown below. In this Figure, the Phase 1 fraction increases up to RH = 60%, and then there isn't much dependence on RH after that. However, the number of samples below 60% is very small.

We tend to think that the previous approach (using 7-day temperature but instantaneous mixing ratio) makes more sense physically. The goal is to use a "background" temperature that is unperturbed by the wave, and it is assumed that the 7-day temperature represent this temperature. Aside from wave anomalies, the basic state of temperature wouldn't vary much over 7 days (in the tropics), so this assumption is probably adequate. For "background" moisture,

ideally we'd be able to get the mixing ratio just before the wave passes through. However, unlike temperature, moisture can vary in short timescales (i.e. due to processes such as convection), so the 7-day average moisture is likely to be biased low compared to the actual background moisture. Because of this, we think that using the time-collocated MLS mixing ratio is a better way to find the relationship between RHib and clouds, although it is by no means ideal. For this reason, we are inclined to keep the previous approach.

[Figure]

Manuscript Changes: At line 305, we added a short discussion on why our RH estimates are high-biased.

P16 line 343: 4-5 km seems larger than the typical wavelength which comes out of your composites Fig. 6

Manuscript changes: We have rewritten this part to be clearer about what we wanted to convey. The new addition is: "The vertical wavelength inferred from the anomalies in our composites is about 3 km. Dzambo et al. (2019) showed that the power spectrum of TTL gravity waves tend to peak at wavelengths of around 4--5 km, though at 3 km there is still considerable power (their Figure 1). These wavelengths are all resolvable by RO, so it can be assumed that this analysis has included a large part of the TTL gravity wave spectrum."

P16 lines 345-348: "Since negative dT0/dz corresponds to a positive cooling rate (due to downward phase propagation as explained by K16), weakened stability, as well as upward vertical motion wave anomalies (according to the gravity wave polarization relationships),": the two points "positive cooling rate" and "upward vertical motion" are equivalent under the usual adiabatic

approximation. One should be removed from the sentence

Manuscript Changes: "positive cooling rate" has been removed here.

**Wording**

P1, line 20: Maybe replace "favor" by "is favorable to" or use a passive.

Manuscript Changes: "favor" changed to "is favorable to"

p7 line 167: 'vertical cloud fraction': I would remove vertical.

Manuscript Changes: "vertical" has been removed

P7 line 168:  'cloud boundaries'  →  'cloud layer'  ?

Manuscript Changes: "boundaries" changed to "layer"

P7 line 175: Again, I would put a passive form.

Manuscript Changes: This paragraph has been rewritten in response to Referee 1's comments, so the word "favor" is no longer there.

P10 line 213 : "a similar compositing technique …"  →  "a compositing technique similar to the one employed above  "

Manuscript Changes: Changed as suggested

P16 l340 : large wavelengths→large horizontal wavelengths

Manuscript Changes: Changed as suggested

**References**

Alexander et al., (2001): Gravity waves in the tropical lower stratosphere: A model study of seasonal and interannual variability,  JGR-atmospheres.

[revised manuscript text omitted]

---

## Referee Report (RR1)

I am pleased with most changes that have been made to the manuscript.
I have two remaining concerns.

One is figure 10.  Per my request, it had been changed it to display four different regions rather than the original altitude layers, but when these new results were inconclusive, the figure was reverted to height layers.  Having looked at the original altitude plot, it seems to me that in terms of the Re distribution, only the highest altitude layer was significantly distinct from the rest.  For the version of this figure that showed geographic regions instead, the authors noted that the 180-120W DJF area had a broader size distribution than did 60E-120E JJA, which was a region characterized by more frequent convection.  The unique characteristics of the other regions were less conclusive.

I would like to see key impacts of altitude and geography combined into one plot.  This could be accomplished different ways, but for example: an upper panel could have all TTL clouds below 16.5 km, the other could be only TTL clouds from 16.5-17.5 km (from the original figure), and the two lower panels could show 180-120W DJF and 60E-120E JJA, respectively, to highlight differences in the distribution (broad versus narrow) with respect to convective activity focused in these areas.  As before, I think it is useful to see the number of samples included within each layer or region.

The other concern came out in my back and forth with Aurelian (the other reviewer) about the numbering of wave phases.  Upon review, I realize the sequence originally came from the K16 paper (specifically their figure 5) and the current manuscript is consistent with theirs in this regard.  While I believe the sequence in K16 is illogical as it stands -- phase 3 and phase 4 results ought to have been swapped – I'm not going to require any change here, for the sake of consistency with K16, and because the adoption of any particular numbering sequence is done by convention.

---

## Author Response (AR2)

Thank you for reviewing our revised manuscript. Below are the reviewers' comments (in black) and our responses (in green).

**Reviewer 1:**

I am pleased with most changes that have been made to the manuscript.

I have two remaining concerns.

One is figure 10. Per my request, it had been changed it to display four different regions rather than the original altitude layers, but when these new results were inconclusive, the figure was reverted to height layers. Having looked at the original altitude plot, it seems to me that in terms of the Re distribution, only the highest altitude layer was significantly distinct from the rest. For the version of this figure that showed geographic regions instead, the authors noted that the 180-120W DJF area had a broader size distribution than did 60E-120E JJA, which was a region characterized by more frequent convection. The unique characteristics of the other regions were less conclusive.

I would like to see key impacts of altitude and geography combined into one plot. This could be accomplished different ways, but for example: an upper panel could have all TTL clouds below 16.5 km, the other could be only TTL clouds from 16.5-17.5 km (from the original figure), and the two lower panels could show 180-120W DJF and 60E-120E JJA, respectively, to highlight differences in the distribution (broad versus narrow) with respect to convective activity focused in these areas. As before, I think it is useful to see the number of samples included within each layer or region.

We have remade Figure 10 to contain clouds in (a) 14.5 to 16.5km, (b) 16.5 to 17.5 km, (c) 120E-180E DJF, and (d) 60E-120E JJA. The latter two are regions likely to be influenced by convection as discussed in the paper. The discussion on r_e (lines 279-296) has been modified according to this new figure.

The other concern came out in my back and forth with Aurelian (the other reviewer) about the numbering of wave phases. Upon review, I realize the sequence originally came from the K16 paper (specifically their figure 5) and the current manuscript is consistent with theirs in this regard. While I believe the sequence in K16 is illogical as it stands -- phase 3 and phase 4 results ought to have been swapped – I'm not going to require any change here, for the sake of consistency with K16, and because the adoption of any particular numbering sequence is done by convention.

Thanks for your comments regarding the naming convention. We agree that would make more sense to swap phase 3 and 4, but we will keep the current convention to facilitate comparison with K16.

**Reviewer 2:**

I am satisfied with the authors response to the reviewers' comments and recommend publication of the paper. I have included below two further formulation suggestions for the authors to consider before final publication.

line 271 'the confinement in Phase 1 may be positioned closer to Phase 2' → 'the region of confinement may overlap with both Phase 1 and 2, with its center still inside Phase 1 but closer to Phase 2.'
Changed as suggested.

line 295: 'were situated closer Phase 2.' → I preferred the initial phrasing. I would suggest something like: '
[revised manuscript text omitted]

---

## Author Response (AR3)

We have made some minor wording and typo edits; the manuscript changes is attached.

[revised manuscript text omitted]